# Efficient Privacy-Preserving Federated Learning With Selective Parameter Encryption

## Abstract

Federated learning trains machine learning models on distributed devices by aggregating local model updates instead of local data. However, privacy concerns arise as aggregating local models on the server may expose sensitive information through inversion attacks. Thus, privacy-preserving methods, such as homomorphic encryption (HE), then become necessary for FL training. However, despite HE's advantages, applying it to FL training suffers from impractical overheads, especially for foundation models. In this paper, we present *an efficient, privacy-preserving federated learning framework that uses selective parameter encryption with theoretical guarantees.* Our approach proposes to selectively encrypt sensitive parameters, significantly reducing both computation and communication overheads during training while providing a quantifiable privacy guarantee. Our framework shows considerable overhead reduction, particularly for large foundation models (e.g. ∼100x reduction for GPT-2), demonstrating its potential for scalable HE-based FL deployment.

## 1 Introduction

Federated learning allows distributed clients to collectively train a global model without directly sharing data. Instead of uploading raw data to a central server for training, clients train models locally and share their model updates with the server, where the model updates are then averaged based on the aggregation functions (McMahan et al., 2017) to obtain a global model. While federated learning ensures that local raw data does not leave their original locations, it remains vulnerable to eavesdroppers and malicious servers that might exploit plaintext model updates to reconstruct sensitive training data (Fig. 1 (left)), i.e., gradient inversion attacks (Zhu et al., 2019; Criswell et al., 2014; Bhowmick et al., 2018; Hitaj et al., 2017; Han et al., 2023; Hatamizadeh et al., 2022; Fowl et al., 2022). This poses a privacy vulnerability especially when local models are trained on small local datasets (e.g., smartphone text data for large language models). Local models derived from these small datasets inherently contain fine-grained information, making it easier for adversaries to extract sensitive information from local models.

Existing defense methods that reduce privacy leakage include differential privacy (DP) (Truex et al., 2019; Byrd & Polychroniadou, 2020) and secure aggregation (Bonawitz et al., 2017; So et al., 2022). DP adds noise to original model updates but may result in model performance degradation due to the privacy noises introduced. On the other hand, secure aggregation employs zero-sum masks to shield local model updates, ensuring that individual updates remain private. However, secure aggregation demands additional interactive synchronization steps and is sensitive to client dropout, making it less practical in real-world FL applications, where the environments of clients are unstable and may face challenges such as unreliable internet connections and software crashes. Compared with the methods above, homomorphic encryption (HE) (Paillier, 1999; Gentry, 2009; Fan & Vercauteren, 2012; Brakerski et al., 2014; Cheon et al., 2017) offers a robust post-quantum secure solution that protects local models against attacks and *provides privacy guarantee while introducing minimal model performance degradation.* As shown in Fig. 1 (middle), HE-based federated learning (FedHE) encrypts local models on clients and performs model aggregation over ciphertexts on the server to protect against privacy attacks, which has been adopted by several FL systems (Roth et al., 2022; IBM, 2022; Zhang et al., 2020; Du et al., 2023) and domain-specific applications (Stripelis et al., 2021; Yao et al., 2023).

Despite these advantages, homomorphic encryption remains a complex cryptographic foundation with significant computation overheads (as shown in Fig. 1 (right)) for real-world FL applications. Prior FedHE solutions mainly employ existing generic HE methods without sufficient optimization for real-world FL deployment (Roth et al., 2022; IBM, 2022; Zhang et al., 2020; Du et al., 2023). The scalability of encrypted computation and communication during federated training then becomes a bottleneck, restricting its feasibility for real-world scenarios. The computation overhead of HE is particularly noticeable, *commonly ∼15x increase in both computation and communication*, both growing linearly w.r.t. the size of models (Cheon et al., 2017; Gouert et al., 2022). Especially across resource-constrained devices, encrypted computing and communication of large models might take considerably longer than the actual model training.

To address these challenges, we propose an efficient homomorphic-encryption-based privacy-preserving FL solution with **Selective Parameter Encryption** for practical deployment[1]. Our method significantly reduces communication and computation overheads, enabling efficient HE-based federated learning. We further provide the first theoretical framework to quantify the privacy guarantee of selective encryption, which indicates a significant improvement over random encryption and differential privacy, with the important observation that most existing models follow Log-Normal Mixture distributions. Extensive experiments validate our privacy quantification framework.

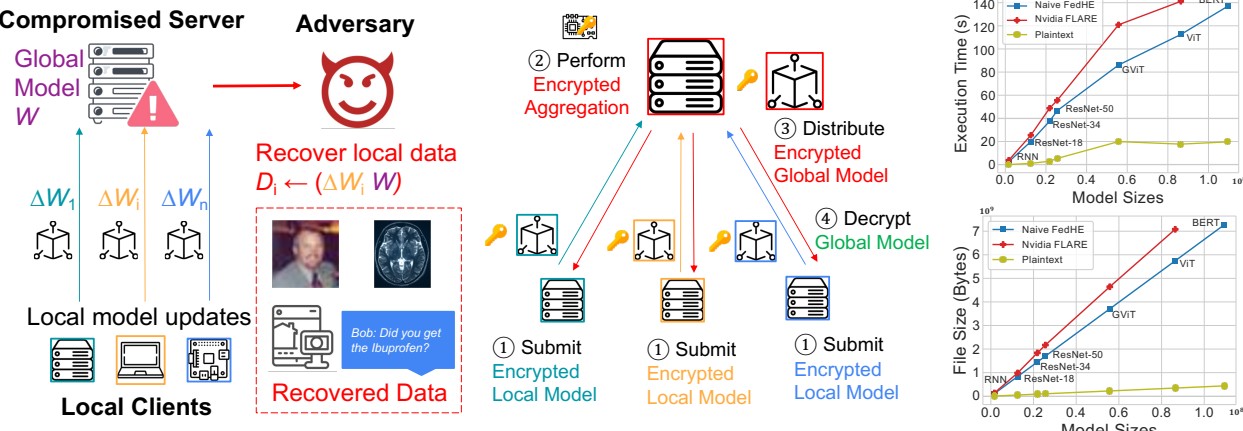

**Figure 1:** (left) Data Reconstruction Attacks: an adversarial server can recover local training data from local model updates and global model at last round; (middle) HE-based Federated Aggregation: models are encrypted and the server acts as a computing service without access to models; (right) Computation and Communication Overhead for Aggregating Fully Encrypted Models: compared with Nvidia Flare (Nvidia, 2021) (which does not have provable selective parameter encryption), overheads include encryption/decryption and encrypted aggregation.

**Key contributions**:

- We propose **Selective Parameter Encryption** in §3 that selectively encrypts the most privacy-sensitive parameters to minimize encrypted model updates and reduce overheads while providing a privacy guarantee quantified by our proposed privacy analysis framework.
- We provide *the theoretical framework* for quantifying the privacy guarantee of selective homomorphic encryption in §4. Selective Parameter Encryption requires significantly less encryption over random selection with provable guarantee validated empirically.
- Extensive experiments in §5 show that the optimized system achieves significant overhead reduction while preserving privacy against state-of-the-art ML privacy attacks, e.g., ∼1000x reduction for ResNet, and ∼100x reduction for GPT-2, demonstrating its potential for real-world deployments.

## 2 Related Work

**Privacy Attacks On FL.** Threats and attacks on privacy in federated learning have been studied in recent years (Mothukuri et al., 2021). Data reconstruction attacks (Criswell et al., 2014; Bhowmick et al., 2018; Hitaj

---

[1]We integrate our work with an open-source federated learning platform.

et al., 2017) exploit local models (or local model updates) to revert sensitive information or even reconstruct the training data. With direct access to more fine-grained local models trained on a smaller dataset (Wang et al., 2019), the adversary can have a higher chance of a successful attack. Moreover, further attacks can be performed using GAN-based attacks to even fully recover the original data (Hitaj et al., 2017). The majority of the privacy attacks can be traced back to the direct exposure of plaintext accesses to local models to other parties.

**Non-HE Defense Mechanisms.** Local differential privacy has been adopted to protect local model updates by adding differential noise on the client side before the server-side aggregation (Truex et al., 2019; Byrd & Polychroniadou, 2020) where privacy guarantee requires large-scale statistical noise on fine-grained local updates that generally degrades model performance (Truex et al., 2020). On the other hand, other work proposes to apply zero-sum masks (usually pair-wise) to mask local model updates such that any individual local update is indistinguishable to the server (Bonawitz et al., 2017; So et al., 2022). However, such a strategy introduces several challenges including key/mask synchronization requirements and federated learner dropouts. Compared to these solutions providing privacy protection in FL, HE is non-interactive and dropout-resilient (vs. general secure aggregation protocols (Bonawitz et al., 2017; So et al., 2022)) and it introduces negligible model performance degradation (vs. noise-based differential privacy solutions (Truex et al., 2019; Byrd & Polychroniadou, 2020)). The comparison of HE vs other privacy-preserving primitives can be found in Table 1.

| | Model Degradation | Overheads | Client Dropout | Interactive Sync |
|---|---|---|---|---|
| Differential Privacy | With noise | **Light** | **Robust** | **No** |
| Secure Aggregation | **Exact** | Medium | Susceptible | Yes |
| Homomorphic Encryption | **Exact** | Heavy | **Robust** | **No** |

**Table 1:** Comparison of Differential Privacy, Secure Aggregation, and Homomorphic Encryption

**Existing HE-based FL Work.** Existing HE-based FL work either apply restricted HE schemes (e.g., additive scheme Paillier) (Zhang et al., 2020; Fang & Qian, 2021; Jiang et al., 2021) without extensibility to further FL aggregation functions or provide a generic but impractical HE implementation on FL aggregation (Jiang et al., 2021; Du et al., 2023; Ma et al., 2022), including industrial platforms such as IBM FL (IBM, 2022), while leaving the key issue with impractical HE overheads as an unresolved question. In our work, we propose a novel Selective Parameter Encryption optimization scheme that largely reduces the overheads as well as provides the first theoretical framework to quantify the privacy guarantee of selective encryption, which makes HE-based FL viable and provable in practical deployments.

**Parameter Selection in ML.** Selective encryption of models has been explored in prior work, particularly in single-client-server machine learning setups for training and inference. For instance, Sphinx (Tian et al., 2022) employs a hybrid approach, utilizing HE for bias parameters while applying DP to the remaining parameters. However, unlike our privacy sensitivity-based method, Sphinx does not easily satisfy the challenges in model and dataset diversity in FL. Similarly, other approaches (Tian et al., 2021) face limitations in FL due to their reliance on specific model architectures, overly coarse layer-wise selection strategies, and the absence of robust privacy quantification.

## 3 Federated Learning With Selective Parameter Encryption

We overview *Selective Parameter Encryption* in FL in §3.1, define the threat model in §3.2, describe the general algorithmic design of HE-based FL in §3.3, and explain how *Selective Parameter Encryption* optimizes the overheads in §3.4.

### 3.1 Methodology Overview

Figure 2 overviews major stages in our efficient HE-based federated training, including *i Encryption key agreement*: the clients generate HE keys using the threshold HE key agreement protocol or trusted key authority; *ii Encryption mask calculation*: the clients and the server apply *Selective Parameter Encryption*

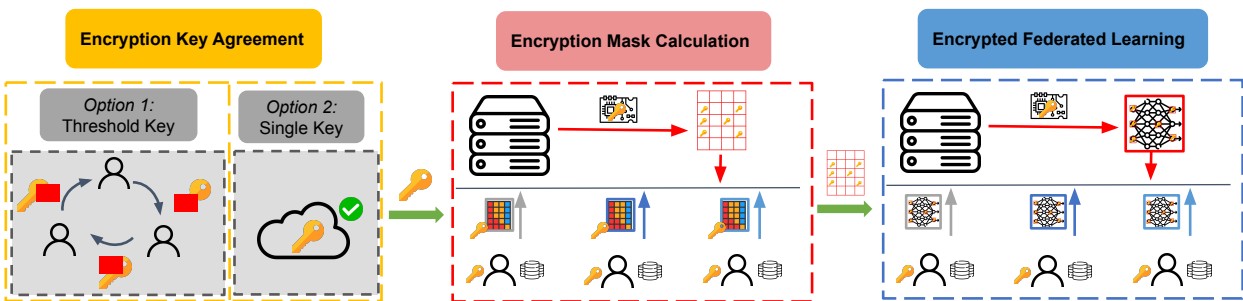

**Figure 2:** Federated Learning Pipeline With Selective Parameter Encryption: in the **Encryption Key Agreement** stage, clients can either use distributed threshold key agreement protocol or outsource a trusted key authority. We simplify the illustration here by abstracting the key pair of the public key and secret key (partial secret keys if using threshold protocol) as one key; in the **Encryption Mask Calculation** stage, clients use local datasets to calculate local model sensitivity maps which are homomorphically aggregated at the server to generate an encryption mask; in the **Encrypted Federated Learning** stage, clients use homomorphic encryption with encryption mask to protect local model updates where the server aggregates them but does not have access to sensitive local models.

to agree on a selective encryption mask; and *iii Encrypted federated learning*: at each round, the clients selectively encrypt local models using the HE key and the encryption mask for efficient encrypted aggregation at the server.

## 3.2 Threat Model

We define a semi-honest adversary $\mathcal{A}$ that can corrupt the aggregation server or any subset of local clients. $\mathcal{A}$ follows the protocol but tries to learn as much information as possible. Loosely speaking, when $\mathcal{A}$ corrupts a subset of clients, the security definition requires that only the private information in local models from the corrupted clients will be learned by $\mathcal{A}$.

When $\mathcal{A}$ corrupts both the aggregation server and a number of clients, the default setup where the private key is shared with all clients (also with corrupted clients) will allow $\mathcal{A}$ to decrypt local models from benign clients (by combining encrypted local models received by the corrupted server and the private key received by any corrupted client). This issue can be mitigated by adopting the threshold or multi-key variant of HE where decryption must be collaboratively performed by a certain number of clients (Aloufi et al., 2021; Ma et al., 2022; Du et al., 2023). Since the multi-key homomorphic encryption issue is not the focus of this work, in the rest of the paper we default to a single-key setup, but provide details on threshold homomorphic encryption federated learning and microbenchmarks in Appendix §A.4.

## 3.3 Algorithm for HE-Based Federated Aggregation

Privacy-preserving federated learning leverages homomorphically encrypted aggregation functions to enable the aggregation server to combine local model parameters without viewing them in their unencrypted form. We primarily focus on FedAvg (McMahan et al., 2017), which has been proved as still one of the best-performing federated aggregation strategies while maintaining computational simplicity (Wang et al., 2022).

Our HE-based secure aggregation algorithm can be summarized as: given an aggregation server and $N$ clients, each client $i \in [N]$ owns a local dataset $D_i$ and initializes a local model $\mathbf{W}_i$ with the aggregation weighing factor $\alpha_i$; the key authority or the distributed threshold key agreement protocol generates a key pair $(pk, sk)$ and the crypto context, then distributes the key pair and crypto context to clients and only the crypto context, which is public, to the server. The clients and the server then collectively calculate a global encryption mask $\mathcal{M}$ for **Selective Parameter Encryption** also using homomorphic encryption. At each round $t \in [T]$, the server performs the aggregation

$$[\mathbf{W}_{\text{glob}}] = \sum_{i=1}^{N} \alpha_i [\![\mathcal{M} \odot \mathbf{W}_i]\!] + \sum_{i=1}^{N} \alpha_i ((\mathbf{1} - \mathcal{M}) \odot \mathbf{W}_i), \tag{1}$$

---

**Algorithm 1** HE-Based Federated Aggregation

---

**Input :** $[\![\mathbf{W}]\!]$: the fully encrypted model;
      $[\mathbf{W}]$: the partially encrypted model;
      $p$: the ratio of parameters for selective encryption;
      $b$: (optional) differential privacy parameter;
**Setup:** // Key Authority Generate Key
      $(pk, sk) \leftarrow HE.KeyGen(\lambda)$;
      // Local Sensitivity Map Calculation
      **for** *each client $i \in [N]$* **do in parallel**
          $\mathbf{W}_i \leftarrow Init(\mathbf{W})$;
          $\mathbf{S}_i \leftarrow Sensitivity(\mathbf{W}, D_i)$;
          $[\![\mathbf{S}_i]\!] \leftarrow Enc(pk, \mathbf{S}_i)$;
          Send $[\![\mathbf{S}_i]\!]$ to server;
      **end**
      // Server Encryption Mask Aggregation
      $[\![\mathcal{M}]\!] \leftarrow Select(\sum_{i=1}^{N} \alpha_i [\![\mathbf{S}_i]\!], p)$;
**Train :**
**for** $t = 1, 2, \dots, T$ **do**
   **for** *each client $i \in [N]$* **do in parallel**
      **if** $t = 1$ **then**
         Receive $[\![\mathcal{M}]\!]$ from server;
         $\mathcal{M} \leftarrow HE.Dec(sk, [\![\mathcal{M}]\!])$;
      **end**
      **if** $t > 1$ **then**
         Receive $[\mathbf{W}_{\text{glob}}]$ from server;
         $\mathbf{W}_i \leftarrow HE.Dec(sk, \mathcal{M} \odot [\mathbf{W}_{\text{glob}}]) + (\mathbf{1} - \mathcal{M}) \odot [\mathbf{W}_{\text{glob}}]$;
      **end**
      $\mathbf{W}_i \leftarrow Train(\mathbf{W}_i, D_i)$;
      // Additional Differential Privacy
      **if** *Add DP* **then**
         $\mathbf{W}_i \leftarrow \mathbf{W}_i + Noise(b)$;
      **end**
      $[\mathbf{W}_i] \leftarrow HE.Enc(pk, \mathcal{M} \odot \mathbf{W}_i) + (\mathbf{1} - \mathcal{M}) \odot \mathbf{W}_i$;
      Send $[\mathbf{W}_i]$ to server $\mathcal{S}$;
   **end**
   // Server Model Aggregation
   $[\mathbf{W}_{\text{glob}}] \leftarrow \sum_{i=1}^{N} \alpha_i [\![\mathcal{M} \odot \mathbf{W}_i]\!] + \sum_{i=1}^{N} \alpha_i ((\mathbf{1} - \mathcal{M}) \odot \mathbf{W}_i)$;
**end**

---

where $[\mathbf{W}_{\text{glob}}]$ is the partially-encrypted global model, $\mathbf{W}_i$ is the $i$-th plaintext local model with $[\![]\!]$ indicating the portion of the model that is fully encrypted, $\alpha_i$ is the aggregation weight for client $i$, and $\mathcal{M}$ is the global model encryption mask (details in Algorithm 1).

We only need one HE multiplicative depth in our algorithm for weighting, which is preferred to reduce HE multiplication operations. Our method can also be easily extended to support more FL aggregation functions with HE by encrypting and computing the new parameters in these algorithms (e.g. FedProx (Li et al., 2020)). We will explain how the encryption mask $\mathcal{M}$ is formalized in the next subsection.

### 3.4 Selective Parameter Encryption

Fully encrypted models can guarantee no access to plaintext local models from the adversary at a cost of high overheads. However, previous work on privacy leakage analysis shows that "partial transparency", e.g. hiding parts of the models (Hatamizadeh et al., 2022; Mo et al., 2020), can limit an adversary's ability to

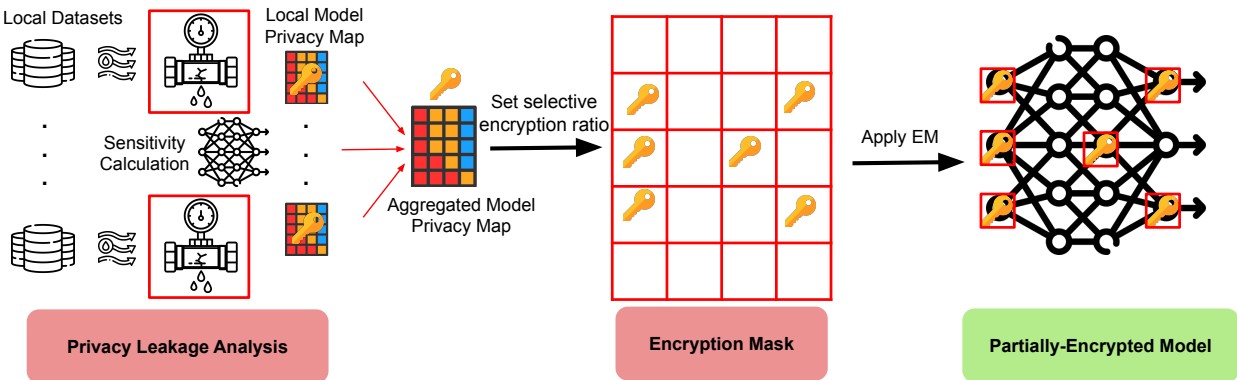

**Figure 3: Selective Parameter Encryption**: in the initialization stage, clients first calculate privacy sensitivities on the model using its own dataset and local sensitivities will be securely aggregated to a global model privacy map. The encryption mask will be then determined by the privacy map and a set selection value $p$ per overhead requirements and privacy guarantee. Only the masked parameters will be aggregated in the encrypted form.

perform privacy attacks like gradient inversion attacks (Lu et al., 2022). Combined with the observation that the overheads of HE are directly related to the size of encrypted model parameters Ma et al. (2022), we propose *Selective Parameter Encryption* to *selectively encrypt the most privacy-sensitive parameters* to reduce overheads while providing quantifiable privacy; see Figure 3.

**Step 1: Privacy Leakage Analysis on Clients.** We adopt sensitivity Novak et al. (2018); Sokolić et al. (2017); Mo et al. (2020) for measuring the general privacy risk on model gradients with respect to the input data. Formally, given model $\mathbf{W}$ and $K$ data samples with input matrix $\mathbf{X}$ and ground truth label vector $\mathbf{y}$, we compute the sensitivity for each parameter $w_m$ as $\frac{1}{K} \sum_{k=1}^{K} \|J_{k,m}\|$, where $J_{k,m}$ can be approximate by the gradient $\frac{\partial^2 \ell(\mathbf{X},\mathbf{y},\mathbf{W})}{\partial x_k \partial w_m}$, $\ell(\cdot)$ is the loss function given $\mathbf{X}$, $\mathbf{y}$ and $\mathbf{W}$, and $\|\cdot\|$ calculates the absolute value. The intuition is to calculate how much the gradient of the parameter will change for each data point $k$. Each client $i$ then sends the encrypted sensitivity $[\![\mathbf{S}_i]\!]$ to the server.

Different parts of a model contribute to attacks by revealing uneven amounts of information. Using this insight, we propose to only select and encrypt parts of the model that are more important and susceptible to attacks to reduce HE overheads while preserving adequate privacy.

**Step 2: Encryption Mask Agreement across Clients.** The sensitivity map is dependent on the model and also the data. With potentially heterogeneous data distributions, the server aggregates local sensitivity maps to a global privacy map $\sum_{i=1}^{N} \alpha_i [\![\mathbf{S}_i]\!]$. The global encryption mask $\mathcal{M}$ is then configured using a privacy-overhead ratio $p \in [0,1]$, i.e., the ratio of selecting the most sensitive parameters for encryption. The global encryption mask is then shared among clients as part of the federated learning configuration.

## 4 Quantifying Privacy Of Selective Parameter Encryption

Although sensitivity calculation provides guidance on selecting important model parameters, to the best of our knowledge there is no existing work that successfully quantifies the privacy guarantee from the model parameter sensitivity. In this section, we provide proof to analyze the privacy guarantee of Selective Parameter Encryption via integrating the theoretical framework of privacy budget analysis (Dwork, 2006).

### 4.1 Encrypted Aggregation Quantified in Privacy Budget

We utilize the differential privacy theory as the basis of privacy guarantee in our approach. Since we adopt selective parameter encryption instead of encrypting the whole model, the information-theoretic differential privacy (Dwork, 2006) cannot be directly applied to our framework. Thus, we adopt a hybrid framework that bounds the computational power of the assumed adversary (Beimel et al., 2008; Vadhan, 2017).

**Definition 4.1** (Computational $(\epsilon|n)$-Differential Privacy (Beimel et al., 2008; Vadhan, 2017))**.** A randomized algorithm $\mathcal{S}$ satisfies computational $(\epsilon|n)$-differential privacy if for any two adjacent datasets $D_1$ and $D_2$ that

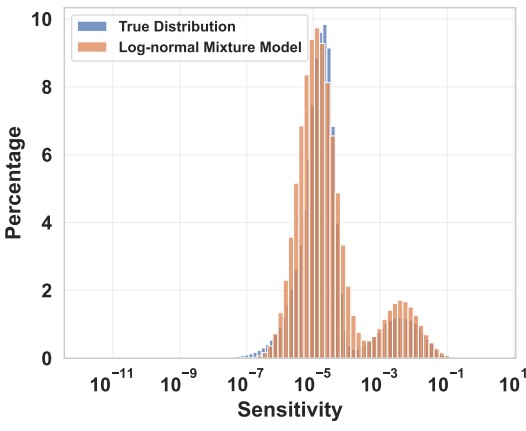
**(a)** Estimation of the Sensitivity Distribution

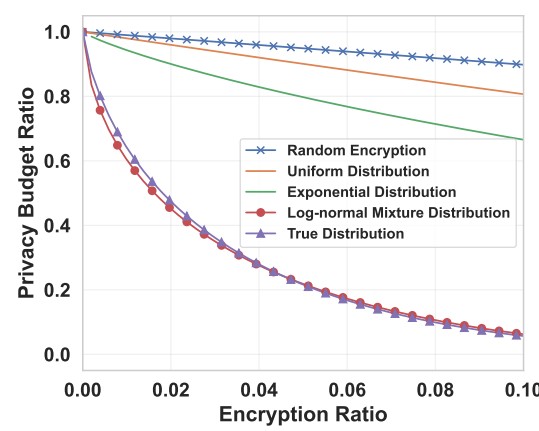
**(b)** Estimation of the Privacy Budget Ratio

**Figure 4:** Sensitivity Distribution and Privacy Budget Ratio from Selective Parameter Encryption (Transformer-3t): calculated parameter sensitivity follows a Log-Normal Mixture distribution, allowing a smaller privacy budget to achieve the same privacy. Similar results on Llama 3.2 can be found in Figure 29.

vary by one data point, and for any possible output $O \subseteq \text{Range}(\mathcal{F})$, given some negligible function $\delta(n)$ for $poly(n)$-bit strings, the following inequality holds[2]:

$$\Pr\left[\mathcal{S}\left(D_1\right) \in O\right] \leq e^{\epsilon} \Pr\left[\mathcal{S}\left(D_2\right) \in O\right] + \delta(n). \tag{2}$$

Computational $(\epsilon|n)$-differential privacy can be achieved by adding the (0-centered) Gaussian mechnism (Dwork et al., 2006a) on model updates. Noises are sampled on a variance $\sigma^2 = \frac{2\Delta f^2 \log(\frac{1.25}{\delta})}{\epsilon^2}$, where non-zero $\delta = \delta(n)$ and $\Delta f$ is the function sensitivity as the maximum difference in the output of a function $f$. $\delta(n)$ is quantified by computational indistinguishability of homomorphic encryption (see Appendix §A.11).

To simplify our quantification process, we adopt the pure DP format by leveraging differential privacy mechanism conversion from approximate DP (Gaussian) to pure DP (Laplace) via zCDP (Bun & Steinke, 2016) (more in Appendix §A.12). The Laplace scale parameter $b$ can be chosen as $b = \frac{\Delta f}{\epsilon}$, such that the Laplace Mechanism satisfies $\epsilon$-privacy.

### 4.2 Selective Parameter Encryption by Privacy Theory

**Lemma 4.2** (Sequential Composition (Dwork et al., 2006b)). *If $\mathcal{S}_1(x)$ satisfies $\epsilon_1$-privacy and $\mathcal{S}_2(x)$ satisfies $\epsilon_2$-privacy, then the mechanism $\mathcal{G}(x) = (\mathcal{S}_1(x), \mathcal{S}_2(x))$ that releases both results satisfies $(\epsilon_1 + \epsilon_2)$-privacy.*

Based on Lemma 4.2, letting $J = \sum_{i=1}^{N} \frac{\Delta f_i}{b}$, we can quantify the privacy of Full DP, random parameter encryption, and Selective Parameter Encryption.

*Remark* 4.3 (Achieving $J$-Privacy by Laplace Mechanism on All Model Parameters). If we add Laplace noise on all parameters with fixed noise scale $b$, it satisfies $J$-privacy.

*Remark* 4.4 (Achieving $(1-p)J$-Privacy by Random Encryption). If we randomly select model parameters with ratio $p$ for homomorphic encryption and add Laplace noise on the remaining parameters, it satisfies $(1-p)J$-privacy.

**Theorem 4.5** ($rJ$-Privacy by Selective Parameter Encryption). *Suppose the sensitivity data follows a distribution with density function $p(x)$, $x \in [0, x_{max}]$. Applying homomorphic encryption on partial model parameters $\Theta$ and Laplace Mechanism on the remaining parameters $[N]/\Theta$ with fixed noise scale $b$ satisfies $rJ$-privacy with the budget ratio*

$$r = \frac{1}{\mu} \int_0^{Q_{1-p}} x p(x) \, dx, \tag{3}$$

---

[2]We omitted the boolean circuit transformation to better align with typical DP definitions.

where $p$ is the fraction of homomorphically encrypted parameters, and $\mu$ and $Q_{1-p}$ are the mean and $(1-p)^{th}$ quantile of $p(x)$ respectively.

The proof of Theorem 4.5 can be found in Appendix §A.13.

*Remark* 4.6. Let $b_0$, $b_1$, and $b_2$ respectively be the scales of Laplace noises necessary for no encryption, (uniform) random encryption, and selective encryption to reach the desired protection level (approximating using $J_0 = J_1 = J_2$). We will have the relation: $b_0 = \frac{1}{1-p}b_1 = \frac{1}{r}b_2$.

Letting $\Delta f \sim \mathcal{D}$, it is clear that the quantification of privacy guarantee from our Selective Parameter Encryption depends on the distribution of the actual parameter distribution $\mathcal{D}$ of a given model.

**Key Observation.** Our extensive experiments indicate that a noticeable collection of popular models' parameters can be closely modeled by the Log-Normal Mixture distribution (as shown by the Transformer-3t example in Fig. 4a and Fig. 4b, with more models in Appendix §A.17). Assuming the sensitivity distribution of a given model follows a Log-normal Mixture distribution $\mathcal{D}'$ ($\mu_i$ as log mean and $\sigma_i$ as log variance), Selective Parameter Encryption requires only $r$ portion of the privacy budget of complete privacy with the same privacy guarantee, where

$$r = \frac{\sum_i \frac{\lambda_i}{\sigma_i} \int_0^{F^{-1}(1-p)} \exp\left(-\frac{(\ln x - \mu_i)^2}{2\sigma_i^2}\right) \mathrm{d}x}{\sqrt{2\pi} \sum_i \lambda_i \exp\left(\mu_i + \frac{\sigma_i^2}{2}\right)}. \tag{4}$$

Compared with random encryption, Selective Parameter Encryption provides much stronger privacy preservation with the same encryption ratio (validated in §5.4). Such a framework can also fit any sensitivity distributions (Uniform and Exponential in Appendix §A.15 and §A.16).

# 5 Evaluation

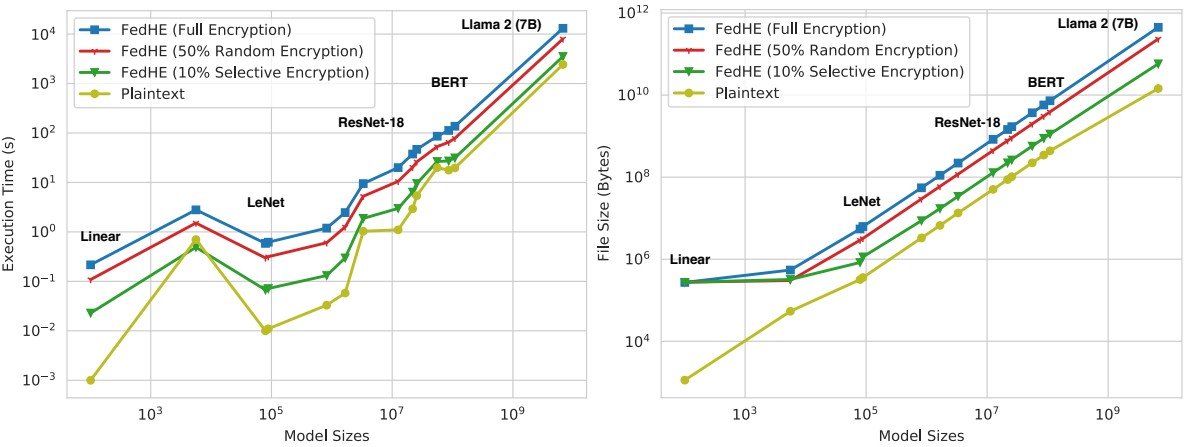

**Figure 5:** Computation (first) and Communication (second) Overhead Comparison For Models of Different Sizes (logarithmic scale): 10% Encryption is based on our selection strategy and 50% encryption is based on random.

In this section, we focus on the evaluation results to show how our proposed Selective Parameter Encryption largely mitigates these overheads for real-world deployment but still guarantees adequate defense against privacy attacks. We also provide the validation of our proposed theoretical privacy quantification. Note that additional experimental results regarding other aspects in FL systems are included in Appendix §A.20.

## 5.1 Experiment Setup

**Models.** We test our framework on models in different ML domains with different sizes including open-source LLMs (more details in Appendix §A.20).

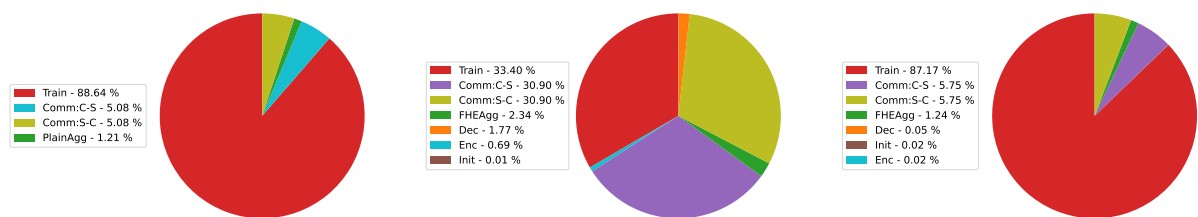

**Figure 6:** Time Distribution of A Training Cycle on ResNet-50 on our industrial deployment platform: plaintext FL (left), HE with full encryption (middle), and HE with selective encryption (right). MLOps test env has a bandwidth of 20 MB/s (Multiple AWS Region). The optimization setup uses an encryption mask with an encrypted ratio $s = 0.01$. Detailed training configuration can be found in Appendix §A.32.

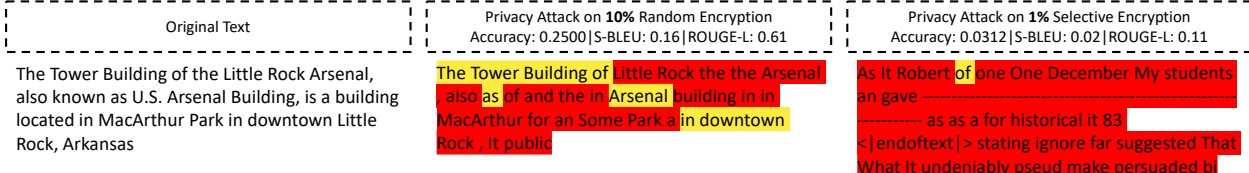

**Figure 7:** Language Model Inversion Attacks (Deng et al., 2021) on GPT-2 with the wikitext Dataset: Red indicates falsely-inverted words and Yellow indicates correctly-inverted words.

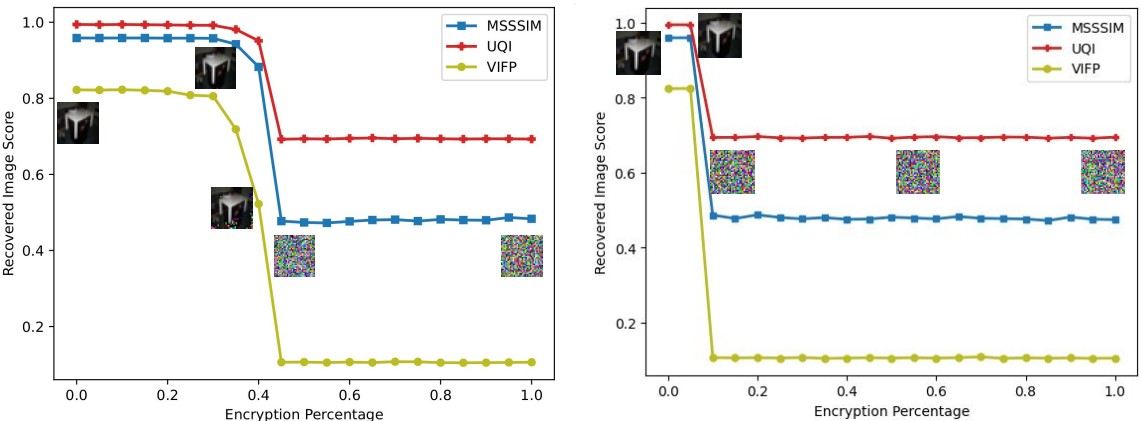

**Figure 8:** Selection Protection Against Gradient Inversion Attack (Zhu et al., 2019) On LeNet with the CIFAR-100 Dataset: attack results when protecting random parameters (first) vs protecting top-$s$ sensitive parameters (second). Each configuration is attacked 10 times and the best-recovered image is selected.

**Attack Dataset.** MNIST dataset ($70k$ images), the CIFAR-100 dataset ($50k$ images), and the WIKITEXT dataset ($100m$ tokens).

**HE Libraries.** We implement our HE core using both PALISADE and TenSEAL. Unless otherwise specified, our results show the evaluation of the PALISADE version.

**Default Crypto Parameters.** Unless otherwise specified, we choose the multiplicative depth of 1, the scaling factor bit digit of 52, an HE packing batch size of 4096, and a security level of 128 as our default HE cryptographic parameters during the evaluation.

**Machines.** (1) For microbenchmarking HE overheads, we use an Intel 8-core 3.60GHz i7-7700 CPU with 32 GB memory and an NVIDIA Tesla T4 GPU; (2) For real MLOps system experiments: we use machines with Intel 6-core 3.70GHz i7-8700K CPU, 64GB memory and NVIDIA GeForce GTX 1080 Ti as clients and an M3 Pro 11-core CPU with 18 GB memory as the aggregation server; (3) For attacking experiments, we use 6 NVIDIA DGX H100 GPUs with 720 GPU hours.

## 5.2 Optimized Overheads

We first examine the overhead optimization gains from Selective Parameter Encryption. Fig. 5 microbenchmarks the overhead reduction from only encrypting certain parts of models, where both overheads are nearly proportional to the size of encrypted model parameters, which is coherent with the general relationship between HE overheads and input sizes. Note that after 10% encryption per our Selective Parameter Encryption, the overheads are close to the ones of plaintext aggregation.

Fig. 6 dissects the training cycle overhead distribution for the HE framework (both with and without optimizations) and the plaintext framework respectively. Note that here we only consider the cost distribution of a single round instead of the entire federated training. This is because, with proper CKKS crypto parameter setup, the model training accuracy of encrypted training has a marginal difference compared to the one of plaintext training even considering the fact that encrypted training has approximate computation under the hood (experimental results regarding this part can be found in Table 5 in Appendix). For a medium-sized model, the majority of overheads (both computation and communication) are shifted to HE-related steps in the full HE mode (w/o optimization) compared to the plaintext mode. However, when optimized by Selective Parameter Encryption, the overheads from HE dramatically drop such that the local training step becomes the majority again.

## 5.3 Effectiveness of Selective Encryption Defense

To evaluate the defense effectiveness of Selective Parameter Encryption, we encrypt model parameters per parameter sensitivity and perform inversion attacks (CV: DLG (Zhu et al., 2019); NLP: TAG (Deng et al., 2021)).

**Defense effectiveness on CV tasks.** We use image samples from CIFAR-100 to calculate the parameter sensitivities of the model. In the DLG attack experiments, we use Multi-scale Structural Similarity Index (MSSSIM), Visual Information Fidelity (VIF), and Universal Quality Image Index (UQI) as metrics to measure the similarity between recovered images and original training images to measure the attack quality hence the privacy leakage. In Fig. 8, compared to random encryption selection where encrypting 42.5% of the parameters can start to protect against attacks, our top-5% encryption selection according to the model privacy map only alone can defend against the attacks, meaning lower overall overhead with the same amount of privacy protection.

**Defense effectiveness on NLP tasks.** We use language samples from the *wikitext* dataset in our experiment. As shown in Fig. 7, with our sensitivity map indicating the top 1% privacy-sensitive parameters, our encryption mask can prevent inversion attacks that yield better defense results than randomly encrypting 10% of the model parameters.

**Empirical Selection Recipe.** In Table 2, we show that empirically, encrypting the top-10% most sensitive parameters tends to be adequate to defend against inversion attacks (Hatamizadeh et al., 2022), but up to 90% are needed for random encryption. We provide the detailed quantitative evaluation in Appendix A.19.

| Model | Size | Selective Encryption | | Random Encryption | |
|---|---|---|---|---|---|
| | | Minimum Encryption Ratio | Attack Score | Minimum Encryption Ratio | Attack Score |
| LeNet | 88,648 | 0.05 | $0.1411 \pm 0.0487$ | 0.11 | $0.1835 \pm 0.0720$ |
| CNN | 2202,660 | 0.001 | $0.1640 \pm 0.0530$ | 0.007 | $0.1861 \pm 0.0494$ |
| ResNet-18 | 11,220,132 | 0.001 | $0.1792 \pm 0.1234$ | 0.05 | $0.1458 \pm 0.0732$ |
| Transformer-3f | 10,702,129 | 0.1 | $0.0000 \pm 0.0000$ | 0.9 | $0.2000 \pm 0.1672$ |
| Transformer-3 | 10,800,433 | 0.1 | $0.0000 \pm 0.0000$ | 0.9 | $0.9750 \pm 0.0415$ |
| Transformer-S | 53,091,409 | 0.1 | $0.0000 \pm 0.0000$ | 0.6 | $0.0875 \pm 0.0573$ |
| GPT-2 | 124,439,808 | 0.01 | $0.0875 \pm 0.0935$ | 0.4 | $0.0644 \pm 0.0720$ |

**Table 2:** Defense Effectiveness on CV and NLP Models: each configuration is attacked 10 times and the best attack score is recorded (VIF for CV tasks and Reconstruction Accuracy for NLP tasks). The minimum encryption ratios are selected as the smallest encryption ratio observed that reduces the attack score to below a certain level (0.2 for VIF of images and 0.1 for Reconstruction Accuracy of texts). The largest encryption ratio used will be recorded if the method fails to provide the desired protection level.

| Model | Enc Ratio | Minimum Laplace Scale | | | $r_1$ | | $r_2$ | |
|---|---|---|---|---|---|---|---|---|
| | | Full DP | Random + DP | Selective + DP | Exp. | Theo. | Exp. | Theo. |
| LeNet | 0.005 | 0.11 | 0.09 | 0.09 | 0.8182 | 0.9950 | 0.8182 | 0.8094 |
| TF-3 | 0.01 | 0.013 | 0.013 | 0.003 | 1.0000 | 0.9995 | 0.2308 | 0.8850 |
| TF-3f | 0.01 | 0.0125 | 0.0125 | 0.0025 | 1.0000 | 0.9999 | 0.2000 | 0.9587 |
| TF-3t | 0.01 | 0.013 | 0.012 | 0.004 | 0.9231 | 0.9990 | 0.3077 | 0.9214 |

**Table 3:** Quantifying Privacy of Selective Parameter Encryption: $r_1$ and $r_2$ represent the ratio of sum induced by the random encryption and selective encryption respectively. The minimum Laplace scales are taken based on the smallest scale of the Laplace noises that reduces the attack score to a desired level. The theoretical value of $r_1$ is one minus the encryption ratio and that of $r_2$ is calculated based on the corresponding sensitivity data.

### 5.4 Privacy Guarantee Quantification

To validate Remark 4.6, we fix the encryption ratio for both random and selective encryption on each selected model and gradually increase the noise scales. When all the encryption methods reach a predefined protection level, we record the minimum noise scale needed and calculate the experimental ratios to make comparison with the theoretical values. The encryption ratio is chosen to be small so that we can observe the influence of the Laplace noises by ensuring the attack score not to be too low at first. As in Table 3, the four cases show with acceptable errors that our theorem provides an upper bound for differential privacy budget of the random and selective encryption methods.

## 6 Conclusion

In this paper, we propose the first practical homomorphic-encryption-based privacy-preserving FL solution with Selective Parameter Encryption to support efficient federated training. Selective Parameter Encryption selectively encrypts the most privacy-sensitive parameters to minimize the size of encrypted model updates to reduce overheads while providing privacy guarantees quantifiable by our proposed theoretical privacy analysis framework. Future work includes: *i*) further improving the performance of threshold HE in the less trusted FL setting; *ii*) investigating the impact of client data heterogeneity Mendieta et al. (2022); Guleria et al. (2024); and *iii*) the potential relationship between explainable ML and our privacy sensitivity calculation.

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

- HE.$KeyGen(\lambda)$: given the security parameter $\lambda$, the key generation algorithm outputs a key pair $(pk, sk)$ and the related cryptographic context.

- HE.$Enc(pk, m)$:the encryption algorithm takes in $pk$ and a plaintext message $m$, then outputs the ciphertext $c$.

- HE.$Eval(c, f)$:the encrypted evaluation algorithm takes in a ciphertext message $c$ and a function $f$, then outputs the computation result $c'$.

- HE.$Dec(sk, c')$:the encryption algorithm takes in $sk$ and a ciphertext message $c'$, then outputs the plaintext $m'$.

**Figure 9:** General Scheme of Homomorphic Encryption

# A    Appendix

## Preliminaries

### A.1    Homomorphic Encryption

Homomorphic Encryption is a cryptographic primitive that allows computation to be performed on encrypted data without revealing the underlying plaintext. It usually serves as a foundation for privacy-preserving outsourcing computing models. HE has generally four algorithms (*KeyGen*, *Enc*, *Eval*, *Dec*) as defined in Figure 9. The fundamental concept is to encrypt data prior to computation, perform the computation on the encrypted data without decryption, and then decrypt the resulting ciphertext to obtain the final plaintext.

Since FL model parameters are usually not integers, our method is built on the Cheon-Kim-Kim-Song (CKKS) scheme (Cheon et al., 2017), a (leveled) HE variant that can work with approximate numbers.

### A.2    Federated Learning

Federated learning is first proposed in (McMahan et al., 2017), which builds distributed machine learning models while keeping personal data on clients. Instead of uploading data to the server for centralized training, clients process their local data and share updated local models with the server. Model parameters from a large population of clients are aggregated by the server and combined to create an improved global model.

The FedAvg (McMahan et al., 2017) is commonly used on the server to combine client updates and produce a new global model. At each round, a global model $\mathbf{W}_{\text{glob}}$ is sent to $N$ client devices. Each client $i$ performs gradient descent on its local data with $E$ local iterations to update the model $\mathbf{W}_i$. The server then does a weighted aggregation of the local models to obtain a new global model, $\mathbf{W}_{\text{glob}} = \sum_{i=1}^{N} \alpha_i \mathbf{W}_i$, where $\alpha_i$ is the weighting factor for client $i$.

Typically, the aggregation runs using plaintext model parameters through a central server (in some cases, via a decentralized protocol), giving the server visibility of each local client's model in plaintext.

## Key Management And Threshold HE

### A.3    HE Key Management

Our general system structure assumes the existence of a potentially compromised aggregation server, which performs the HE-based secure aggregation. Alongside this aggregation server, there also exists a trusted key authority server that generates and distributes HE keys and related crypto context files to authenticated parties (as described previously in Algorithm 1 in the main paper. We assume there is no collusion between these two servers.

Moreover, secure computation protocols for more decentralized settings without an aggregation server are also available using cryptographic primitives such as Threshold HE (Aloufi et al., 2021), Multi-Key HE (Aloufi et al., 2021), and Proxy Re-Encryption (Ateniese et al., 2006; Jin et al., 2022). In such settings, secure computation and decryption can be collaboratively performed across multiple parties without the need for a centralized point. We plan to introduce a more decentralized version in the future. Due to the collaborative nature of such secure computation, the key management will act more as a coordination point instead of a trusted source for key generation.

### A.4 FL With Threshold HE

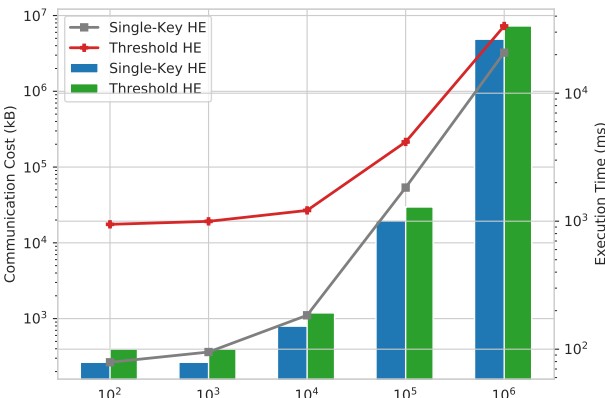

**Figure 10:** Microbenchmark of Threshold-HE-Based FedAvg Implementation: with the x-axis showing the sizes of vectors being aggregated, we use a two-party threshold setup. Both the single-key variant and the threshold variant are configured with an estimated precision of 36 for a fair comparison. Note that bars represent communication overheads and lines represent computation overheads.

The threshold variant of HE schemes is generally based on Shamir's secret sharing (Shamir, 1979) (which is also implemented in PALISADE). Key generation/agreement and decryption processes are in an interactive fashion where each party shares partial responsibility for the task. Threshold key generation results in each party holding a share of the secret key and threshold decryption requires each party to partially decrypt the final ciphertext result and merge to get the final plaintext result. We provide benchmarkings of the threshold-HE-based FedAvg implementation in Figure 10.

## Framework and Platform Deployment

### A.5 Software Framework: Homomorphic Encryption In Federated Learning

In this part, we will illustrate how we design our HE-based aggregation from a software framework perspective.

Figure 11 provides a high-level design of our framework, which consists of three major layers:

- **Crypto Foundation.** The foundation layer is where Python wrappers are built to realize HE functions including key generation, encryption/decryption, secure aggregation, and ciphertext serialization using open-sourced HE libraries;
- **ML Bridge.** The bridging layer connects the FL system orchestration and cryptographic functions. Specifically, we have ML processing APIs to process inputs to HE functions from local training processes and outputs vice versa. Additionally, we realize the optimization module here to mitigate the HE overheads;
- **FL Orchestration.** The FL system layer is where the key authority server manages the key distribution and the (server/client) managers and task executors orchestrate participants.

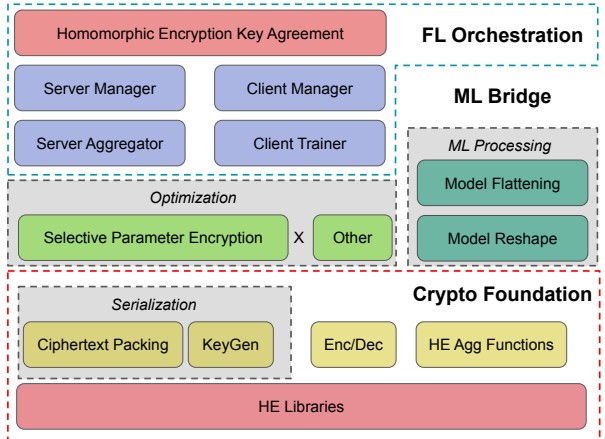

**Figure 11:** Framework Structure: our framework consists of a three-layer structure including Crypto Foundation to support basic HE building blocks, ML Bridge to connect crypto tools with ML functions, and FL Orchestration to coordinate different parties during a task.

Our layered design makes the HE crypto foundation and the optimization module *semi-independent*, allowing different HE libraries to be easily switched in our system and further FL optimization techniques to be easily added to the system.

## A.6    Framework APIs

Table 4 shows the framework APIs in our system related to HE.

| API Name | Description |
|---|---|
| $pk,\ sk = $ **key\_gen**($params$) | Generate a pair of HE keys (public key and private key) |
| $1d\_local\_model = $ **flatten**($local\_model$) | Flatten local trained model tensors into a 1D local model |
| $enc\_local\_model = $ **enc**($pk,\ 1d\_model$) | Encrypt the 1D model |
| $enc\_global\_model = $ **he\_aggregate**( $enc\_models[n],\ weight\_factors[n]$) | Homomorphically aggregate a list of 1D local models |
| $dec\_global\_model = $ **dec**($sk,\ enc\_global\_model$) | Decrypt the 1D global model |
| $global\_model = $ **reshape**( $dec\_global\_model,\ model\_shape$) | Reshape the 1D global model back to the original shape |

**Table 4:** HE Framework APIs

## A.7    Deploy Anywhere: A Deployment Platform MLOps For Edges/Cloud

We implement our deployment-friendly platform such that our system can be easily deployed across cloud and edge devices. Before the training starts, a user uploads the configured server package and the local client package to the web platform. The server package defines the operations on the FL server, such as the aggregation function and client sampling function; the local client package defines the customized model architecture to be trained (model files will be distributed to edge devices in the first round of the training). Both packages are written in Python. The platform then builds and runs the docker image with the uploaded server package to operate as the server for the training with edge devices configured using the client package.

As shown in Figure 12, during the training, users can also keep tracking the learning procedure including device status, training progress/model performance, and system overheads (e.g., training time, communication time, CPU/GPU utilization, and memory utilization) via the web interface. Our platform keeps close track of overheads, which allows users to in real-time pinpoint HE overhead bottlenecks if any.

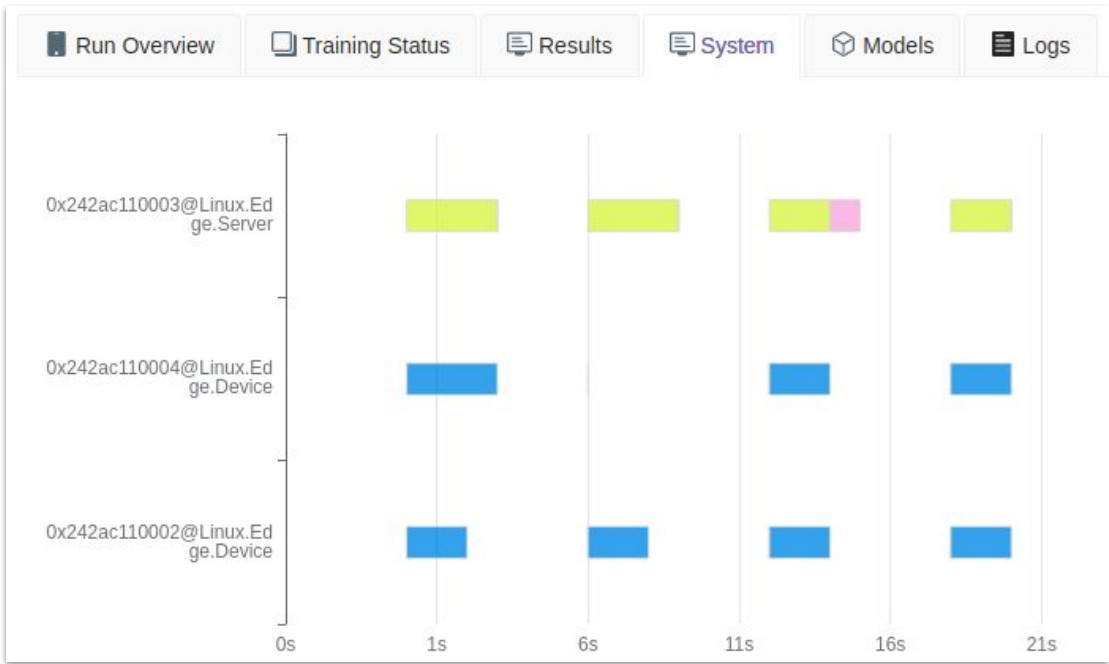

**Figure 12:** Deployment Interface Example: Overhead distribution monitoring on each edge device (e.g. Desktop (Ubuntu), Laptop (MacBook), and Raspberry Pi 4), which can be used to pinpoint HE overhead bottlenecks and guide optimization.

## Additional Definitions And Proofs

### A.8  Adjacent Datasets

**Definition A.1** (Adjacent Datasets)**.** Two datasets $D_1$ and $D_2$ are said to be adjacent if they differ in the data of exactly one individual. Formally, they are adjacent if:

$$|D_1 \Delta D_2| = 1$$

### A.9  Laplace Mechanism

**Definition A.2** (Laplace mechanism)**.** Given a function $f : \mathcal{D} \to \mathbb{R}$,

where $\mathcal{D}$ is the domain of the dataset and $d$ is the dimension of the output, the Laplace mechanism adds Laplace noise to the output of $f$.

Let $b$ be the scale parameter of the Laplace distribution, which is given by:

$$\text{Lap}(x \mid b) = \frac{1}{2b} e^{-\frac{|x|}{b}}$$

Given a dataset $D$, the Laplace mechanism $\mathcal{F}$ is defined as:

$$\mathcal{M}(D) = f(D) + \text{Lap}(0 \mid b)^d$$

### A.10  Differential Privacy Sensitivity

**Definition A.3** (Differential Privacy Sensitivity)**.** To ensure $\epsilon$-privacy, we need to determine the appropriate scale parameter $b$. The DP sensitivity $\Delta f$ of a function $f$ is the maximum difference in the output of $f$ when applied to any two adjacent datasets:

$$\Delta f = \max_{D_1, D_2 : |D_1 \Delta D_2| = 1} \| f(D_1) - f(D_2) \|_1 .$$

**Definition A.4** (Gradient-Based Sensitivity). For a function $f : \mathbb{R}^n \mapsto \mathbb{R}$, its gradient-based sensitivity $\Delta f \in \mathbb{R}^n$ can be evaluated as its gradient

$$\Delta f = \frac{\partial f(D)}{\partial D}.$$

As mentioned by Section 3.4, we adopt the gradient of $f$ as sensitivity (see Definition A.4) which appears to be different from the form in Definition A.3. However, we argue that this notion is loosely compatible with the use of differential privacy if we view it as an extension to the continuous case, i.e., $|D_1 - D_2| = 1$ is replaced with $|D_1 - D_2| \leq \varepsilon$ for some small $\varepsilon$.

## A.11 Proof of Base Full Encryption Protocol

In this subsection, we prove the privacy of base protocol where homomorphic-encryption-based federated learning utilizes the full model parameter encryption (i.e., the selective parameter encryption rate is set to be $1$). We define the adversary in Definition A.5 and privacy in Definition A.7.

**Definition A.5** (Single-Key Adversary). *A semi-honest adversary $\mathcal{A}$ can corrupt (at the same time) any subset of $n$ learners and the aggregation server, but not at the same time.*

Note that the ref of the proof assumes the single-key setup and the privacy of the threshold variant of HE-FL (as shown in Definition A.6) can be easily proved by extending the proofs of threshold homomorphic encryption (Boneh et al., 2006; Laud & Ngo, 2008; Asharov et al., 2012).

**Definition A.6** (Threshold Adversary). *A semi-honest adversary $\mathcal{A}_{\mathcal{T}}\langle$ can corrupt (at the same time) any subset of $n - k$ learners and the aggregation server.*

**Definition A.7** (Privacy). *A homomorphic-encryption federated learning protocol $\pi$ is simulation secure in the presence of a semi-honest adversary $\mathcal{A}$, there exists a simulator $\mathcal{S}$ in the ideal world that also corrupts the same set of parties and produces an output identically distributed to $\mathcal{A}$'s output in the real world.*

**Ideal World.** Our ideal world functionality $\mathcal{F}$ interacts with learners and the aggregation server as follows:

- Each learner sends a registration message to $\mathcal{F}$ for a federated training model task $\mathbf{W}_{\text{glob}}$. $\mathcal{F}$ determines a subset $N' \subset N$ of learners whose data can be used to compute the global model $\mathbf{W}_{\text{glob}}$.

- Both honest and corrupted learners upload their local models to $\mathcal{F}$.

- If local models $\vec{\mathbf{W}}$ of learners in $N'$ are enough to compute $\mathbf{W}_{\text{glob}}$, $\mathcal{F}$ sends $\mathbf{W}_{\text{glob}} \leftarrow \sum_{i=1}^{N'} \alpha_i \mathbf{W}_i$ to all learners in $N'$, otherwise $\mathcal{F}$ sends empty message $\perp$.

**Real World.** In real world, $\mathcal{F}$ is replaced by our protocol described in Algorithm 1 with full model parameter encryption.

We describe a simulator $\mathcal{S}$ that simulates the view of the $\mathcal{A}$ in the real-world execution of our protocol. Our privacy definition A.7 and the simulator $\mathcal{S}$ prove both confidentiality and correctness. We omit the simulation of the view of $\mathcal{A}$ that corrupts the aggregation server here since the learners will not receive the ciphertexts of other learners' local models in the execution of $\pi$ thus such a simulation is immediate and trivial.

**Simulator.** In the ideal world, $\mathcal{S}$ receives $\lambda$ and $1^n$ from $\mathcal{F}$ and executes the following steps:

1. $\mathcal{S}$ chooses a uniformly distributed random tape $r$.

2. $\mathcal{S}$ runs the key generation function to sample $pk$: $(pk, sk) \leftarrow HE.KeyGen(\lambda)$.

3. For a chosen $i$th learner, $\mathcal{S}$ runs the encryption function to sample: $(c_i) \leftarrow HE.Enc(pk, r^{|\mathbf{W}_i|})$.

4. $\mathcal{S}$ repeats Step 3 for all other learners to obtain $\vec{c}$, and runs the federated aggregation function $f$ to sample: $(c_{\text{glob}}) \leftarrow HE.Eval(\vec{\mathbf{c}}, f)$.

The execution of $\mathcal{S}$ implies that:

$$\{(c_i, c_{\text{glob}})\} \stackrel{\text{s}}{\equiv} \left\{ \left( HE.Enc(pk, \mathbf{W}_i), HE.Eval(\vec{\mathbf{W}}, f) \right) \right\}$$

Thus, we conclude that $\mathcal{S}$'s output in the ideal world is computationally indistinguishable from the view of $\mathcal{A}$ in a real world execution:

$$\{\mathcal{S}\left(1^n, (\lambda)\right)\} \overset{\text{s}}{\equiv} \{\text{view}^\pi\left(\lambda\right)\},$$

where view is the view of $\mathcal{A}$ in the real execution of $\pi$.

### A.12   Quantifying negligible privacy value in full encryption

Given a security parameter $\lambda$ that denotes the desired security level of the scheme, i.e., $\lambda$-bit security, we can obtain a relaxed catastrophic failure probability $\delta_0 = \frac{1}{2^\lambda}$, which satisfies $(\epsilon_{approx}, \delta_0)$-DP under approximate DP (Gaussian mechanism), where $\epsilon_{approx} = 0$. Note that, in general for approximate DP, the Gaussian mechanism will not actually release the entire dataset under catastrophic failure probability, rather it fails gracefully, thus $\delta_0$ is a good approximation of the catastrophic failure probability under the failure of the security scheme.

With $(\epsilon_{approx}, \delta_0)$-DP, we can switch the pure DP we used in our paper to approximate DP and use Advanced Composition (Dwork et al., 2010) (Theorem 3.20) to get a tight composition. On the other hand, to compose the privacy of $(\epsilon_{approx}, \delta_0)$-DP under the Gaussian mechanism into our current pure DP composition in the paper, we can also use Lemma 3.7 (Bun & Steinke, 2016) to obtain a partial converse (up to a loss in parameters) from approximate DP to pure DP via zCDP:
With

$$\delta_0 = \frac{1}{2^\lambda}, \tag{5}$$

$$\rho = \epsilon_{approx} + 2\ln\frac{1}{\delta_0} - 2\sqrt{\ln\frac{1}{\delta_0}(\epsilon_{approx} + \ln\frac{1}{\delta_0})}, \tag{6}$$

$$\epsilon_0 = \sqrt{2\rho}, \tag{7}$$

we can have $\epsilon_0 = \sqrt{2\epsilon_{approx} + 2\ln\frac{1}{2^\lambda} - 2\sqrt{\ln\frac{1}{2^\lambda}(\epsilon_{approx} + \ln\frac{1}{2^\lambda})}}$.

Let $\epsilon_{approx} = 10^{-12}$ and $\lambda = 128$ for 128-bit security, we can have a negligible $\epsilon_0 = 9.97 * 10^{-07}$. Note that $\epsilon_{approx} = 10^{-12}$ is a really conservative value for estimating privacy from encryption, when $\epsilon_{approx} = 0$ we can have $\epsilon_0 \simeq 0$. Thus, we have $\epsilon_0$-DP from security of encryption, where $\epsilon_0 \simeq 0$.

### A.13   Proof of $rJ$-Privacy by Selective Parameter Encryption

*Proof.* The mean value of sensitivity within $[0, Q_{1-p}]$ is calculated by

$$E[X|X \leq Q_{1-p}] = \frac{1}{1-p}\int_0^{Q_{1-p}} xp(x)\mathrm{d}x.$$

Suppose the total number of parameters is $n$, the ratio is then obtained as

$$r = \frac{n(1-p)\frac{1}{1-p}\int_0^{Q_{1-p}} xp(x)\mathrm{d}x}{n\mu} = \frac{1}{\mu}\int_0^{Q_{1-p}} xp(x)\mathrm{d}x.$$

Therefore, the total privacy budget is

$$J' = \sum_{i\in[N]/\mathcal{S}} \frac{\Delta f_i}{b} = r\sum_{i=1}^{N} \frac{\Delta f_i}{b} = rJ.$$

$\square$

### A.14 Proof of Privacy Budget Relationship Under Different Parameter Encryption Options

*Proof.* $b_m$ induces the privacy budget of $\varepsilon_i^{(m)} = \frac{\Delta f_i}{b_m}$ for the encryption method indicated by $m$. The total privacy budgets for all the methods are then given by

$$J_0 = \sum_i \varepsilon_i^{(0)} = \frac{1}{b_0} \sum_i \Delta f_i,$$

$$J_1 = (1-p) \sum_i \varepsilon_i^{(1)} = \frac{1-p}{b_1} \sum_i \Delta f_i,$$

$$J_2 = r \sum_i \varepsilon_i^{(2)} = \frac{r}{b_2} \sum_i \Delta f_i.$$

When the methods reach a similar protection level (approximating using $J_0 = J_1 = J_2$), we have the relation above by canceling out the term $\sum_i \Delta f_i$. □

### A.15 Selective Parameter Encryption Privacy Proof Under Uniform Distribution

Assume $\Delta f \sim \mathcal{U}(0,1)$ where $\mathcal{U}$ represents the uniform distribution, we can have the following privacy quantification.

*Remark* A.8 (Achieving $(1-p)^2 J$-Privacy by Sensitive Parameter Selection (Uniformly Distributed Sensitivity)). If we select the most sensitive parameters with ratio $p$ for homomorphic encryption and add Laplace noise on remaining parameters, it satisfies $(1-p)^2 J$-Privacy.

*Proof.* For a uniform distribution with density function $p(x) = \frac{1}{x_{max}}$, $x \in [0, x_{max}]$, mean $\mu = \frac{1}{2} x_{max}$, and $(1-p)$th quantile $Q_{1-p} = (1-p) x_{max}$,

$$r = \frac{2}{x_{max}} \int_0^{(1-p)x_{max}} \frac{x}{x_{max}} \mathrm{d}x = (1-p)^2.$$

□

Uniform distribution is a conservative estimation of the sensitivity distribution. In our experiments, the obtained sensitivity data is mostly right-skewed and can be well modeled by the mixture of several log-normal distributions (see the case of Transformer-3 as shown in Figure 13). However, it is hard to analytically depict the conclusion for log-normal distributions, so we provide Remark A.9 as a demonstration of the right-skewed case with the simpler exponential distribution.

### A.16 Selective Parameter Encryption Privacy Proof Under Exponential Distribution

*Remark* A.9 (Achieving $(p \ln p - p + 1) J$-Privacy by Sensitive Parameter Encryption (Exponentially Distributed Sensitivity)).

*Proof.* For an exponential distribution with density function $p(x) = \lambda e^{-\lambda x}$, mean $\mu = \frac{1}{\lambda}$, and $(1-p)$th quantile $Q_{1-p} = -\frac{\ln p}{\lambda}$. The corresponding ratio is then

$$r = \lambda \int_0^{-\frac{\ln p}{\lambda}} \lambda x e^{-\lambda x} \mathrm{d}x = p \ln p - p + 1.$$

□

Taking Transformer-3t as an example, the estimated privacy budget ratio for sensitivity data under different distributions is presented in Figure 4b. It is clear from the figure that a better fitting of the sensitivity data distribution yields a better estimation of the privacy budget ratio. Note that the estimation here is imperfect since finding the best fitting is not the main concern of our study, but is sufficient to show the correctness of our theorem.

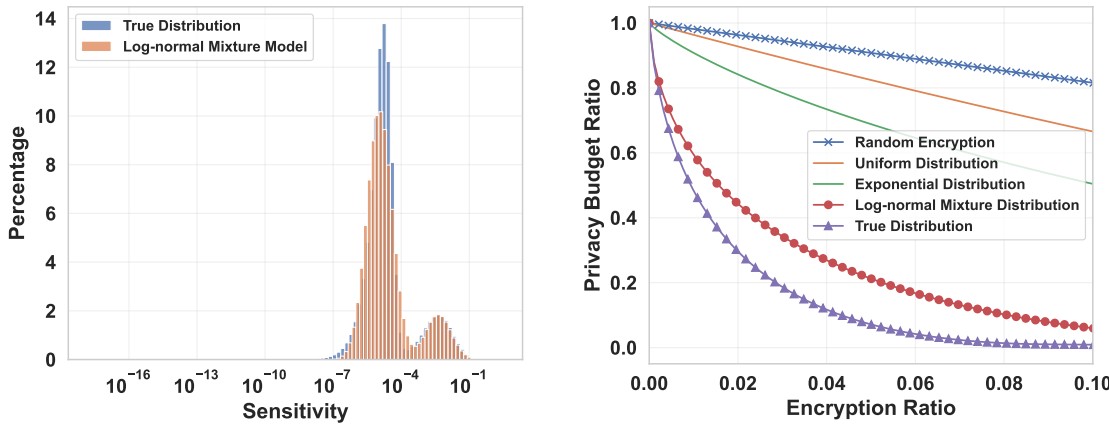

**(a)** Estimation of the Sensitivity Distribution  **(b)** Estimation of the Privacy Budget Ratio

**Figure 13:** Sensitivity Distribution and Privacy Budget Ratio from Selective Parameter Encryption (Transformer-3).

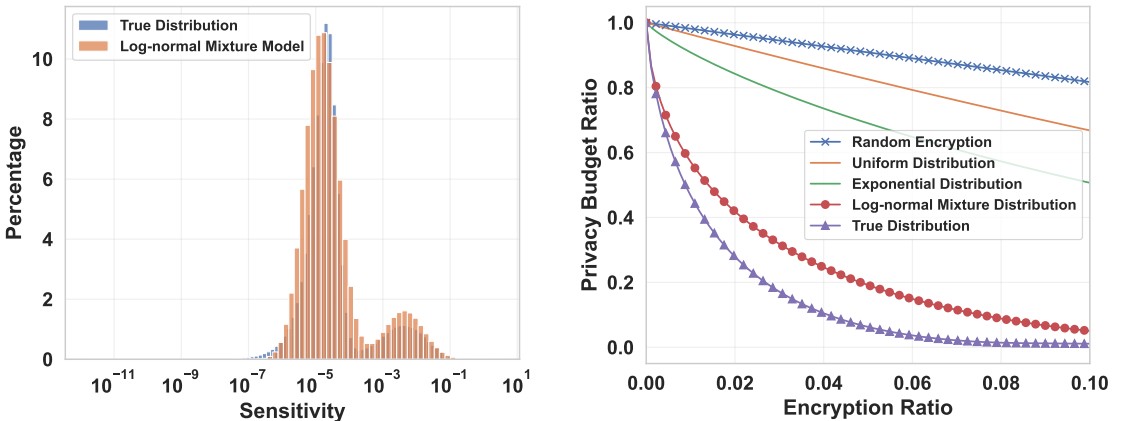

**(a)** Estimation of the Sensitivity Distribution  **(b)** Estimation of the Privacy Budget Ratio

**Figure 14:** Sensitivity Distribution and Privacy Budget Ratio from Selective Parameter Encryption (Transformer-3f).

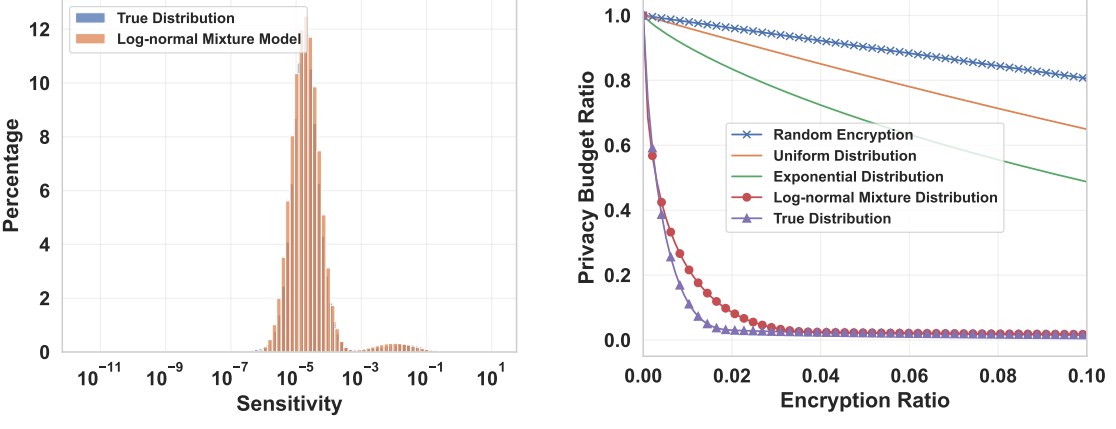

**(a)** Estimation of the Sensitivity Distribution  **(b)** Estimation of the Privacy Budget Ratio

**Figure 15:** Sensitivity Distribution and Privacy Budget Ratio from Selective Parameter Encryption (Transformer-S).

## A.17 Sensitivity Distribution and Privacy Budget Ratio of the Models Included

Figure 13, 14, 15, 16, 17, 18, and 19 show that the log-normal mixture model is a good fitting on the models we use for our evaluation experiments.

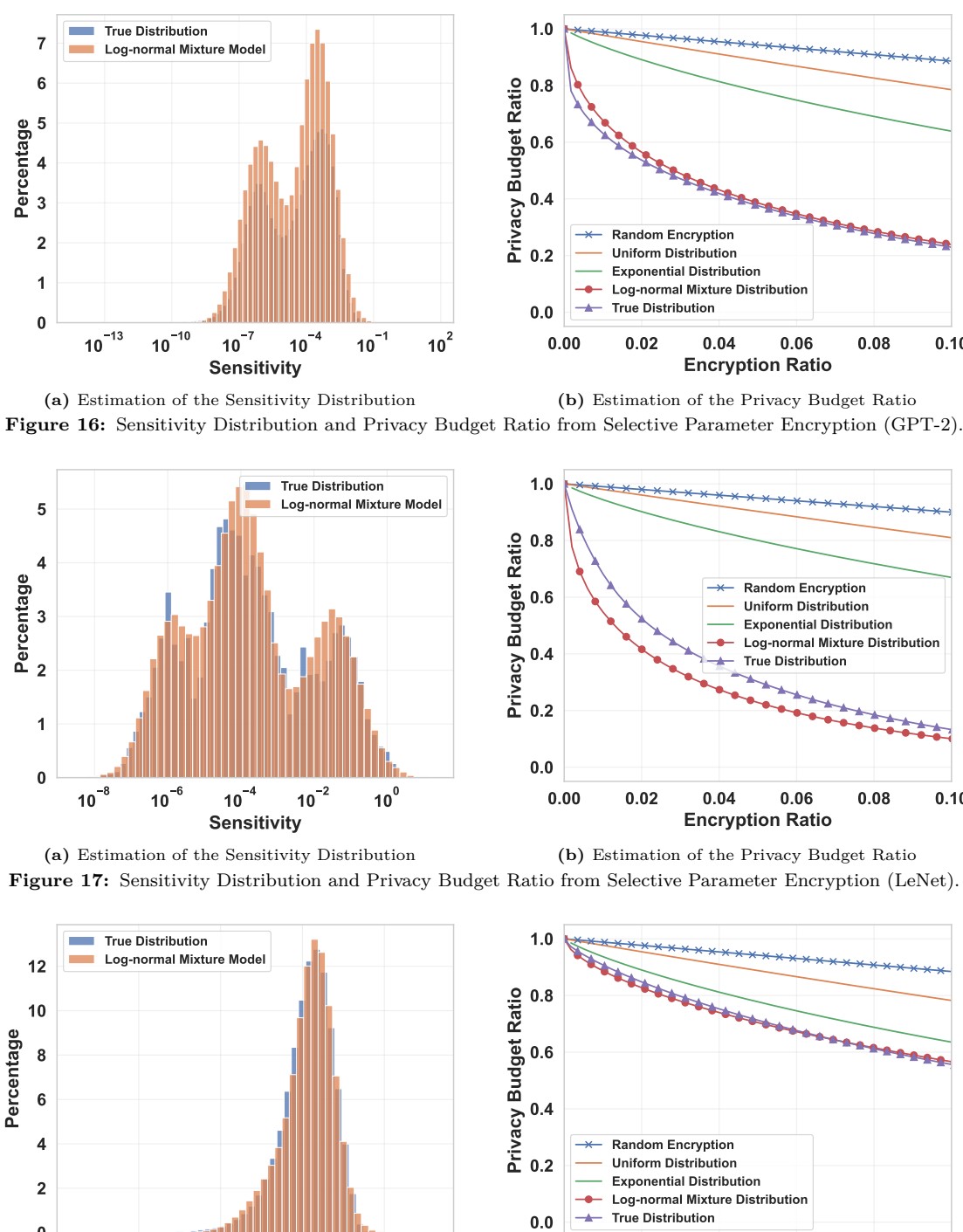

(a) Estimation of the Sensitivity Distribution    (b) Estimation of the Privacy Budget Ratio

**Figure 16:** Sensitivity Distribution and Privacy Budget Ratio from Selective Parameter Encryption (GPT-2).

(a) Estimation of the Sensitivity Distribution    (b) Estimation of the Privacy Budget Ratio

**Figure 17:** Sensitivity Distribution and Privacy Budget Ratio from Selective Parameter Encryption (LeNet).

(a) Estimation of the Sensitivity Distribution    (b) Estimation of the Privacy Budget Ratio

**Figure 18:** Sensitivity Distribution and Privacy Budget Ratio from Selective Parameter Encryption (CNN).

## Supporting Materials for Defense Effectiveness Experiments

### A.18    Parameter Sensitivity Map for LeNet

Figure 20 visualizes the parameter sensitivity map of LeNet.

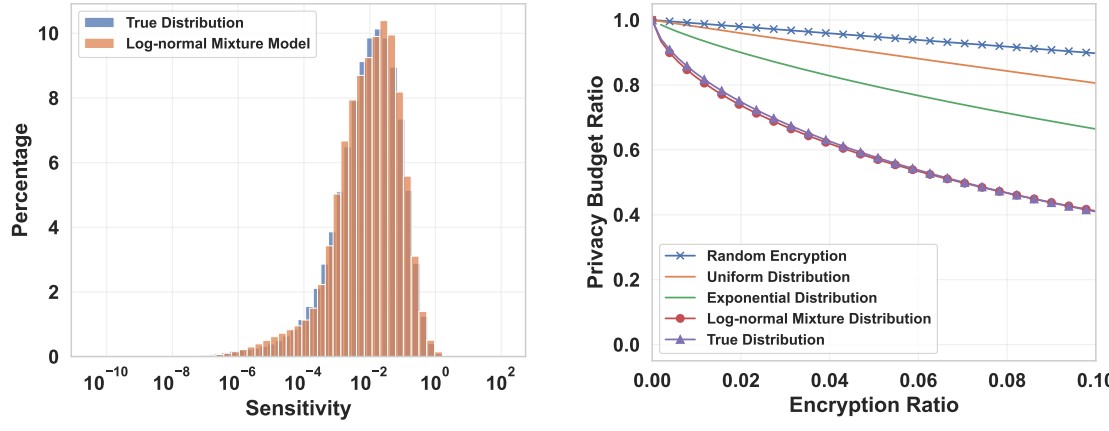

(a) Estimation of the Sensitivity Distribution · (b) Estimation of the Privacy Budget Ratio

**Figure 19:** Sensitivity Distribution and Privacy Budget Ratio from Selective Parameter Encryption (ResNet-18).

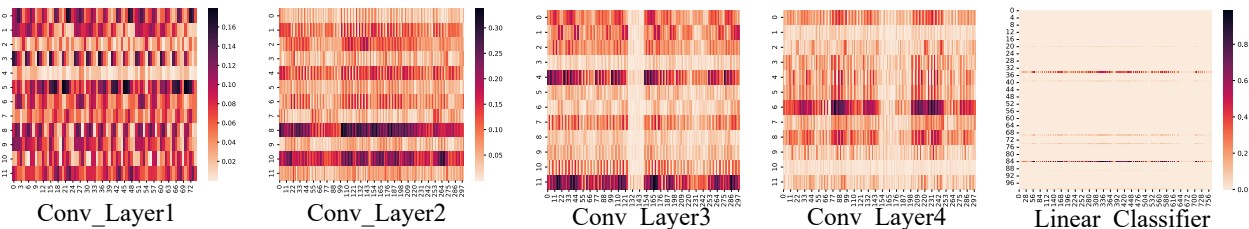

Conv_Layer1 · Conv_Layer2 · Conv_Layer3 · Conv_Layer4 · Linear_Classifier

**Figure 20:** Model Privacy Map Calculated by Sensitivity on LeNet: darker color indicates higher sensitivity. Each subfigure shows the sensitivity of parameters of the current layer. The sensitivity of parameters is imbalanced and many parameters have very little sensitivity (its gradient is hard to be affected by tuning the data input for attack).

### A.19 Defense Effectiveness on CV and NLP Models

Figure 21 and 22 are used for the records of Table 2.

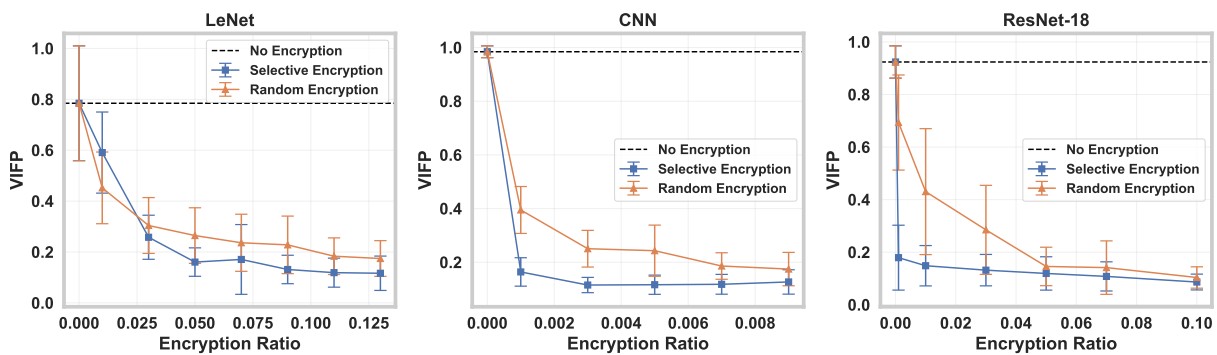

**Figure 21:** Results for Selected CV Models

### A.20 Experiments on Quantifying Privacy

Figure 23 shows the privacy guarantee of Selective Parameter Encryption using the equivalent privacy budget.

## Additional Experiments

We evaluate the HE-based training overheads (without our optimization in place) across various FL training scenarios and configurations. This analysis covers diverse model scales, HE cryptographic parameter configurations, client quantities involved in the task, and communication bandwidths. This helps us to

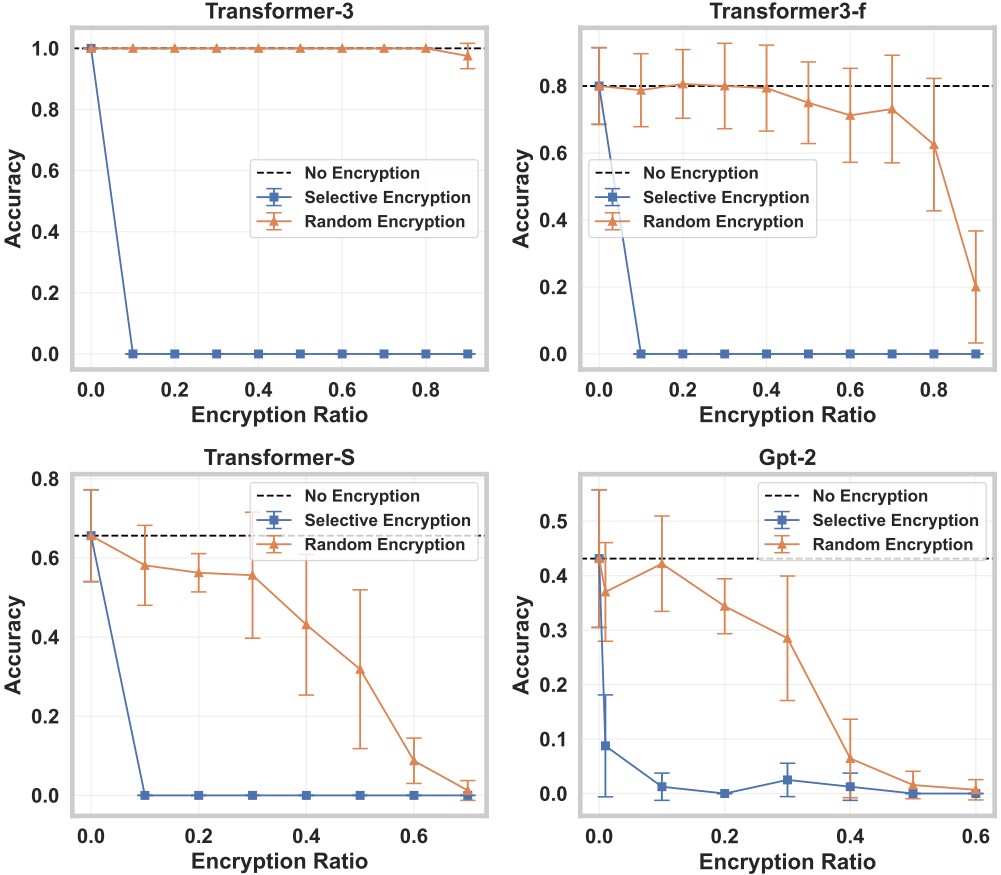

**Figure 22:** Results for Selected NLP Models

identify bottlenecks in the HE process throughout the entire training cycle. We also benchmark our framework against other open-source HE solutions to demonstrate its advantages.

## A.21 Parameter Efficiency Techniques in HE-Based FL

Table 6 shows the optimization gains by applying model parameter efficiency solutions in HE-Based FL.

## A.22 Results on Different Scales of Models

We evaluate our framework on models with different size scales and different domains, from small models like the linear model to large foundation models such as Vision Transformer (Dosovitskiy et al., 2020) and BERT (Devlin et al., 2018). As Table 5 show, both computational and communicational overheads are generally proportional to model sizes.

Table 5 illustrates more clearly the overhead increase from the plaintext federated aggregation. The computation fold ratio is in general 5x ~ 20x while the communication overhead can jump to a common 15x. Small models tend to have a higher computational overhead ratio increase. This is mainly due to the standard HE initialization process, which plays a more significant role when compared to the plaintext cost. The communication cost increase is significant for models with sizes smaller than 4096 (the packing batch size) numbers. Recall that the way our HE core packs encrypted numbers makes an array whose size is smaller than the packing batch size still requires a full ciphertext.

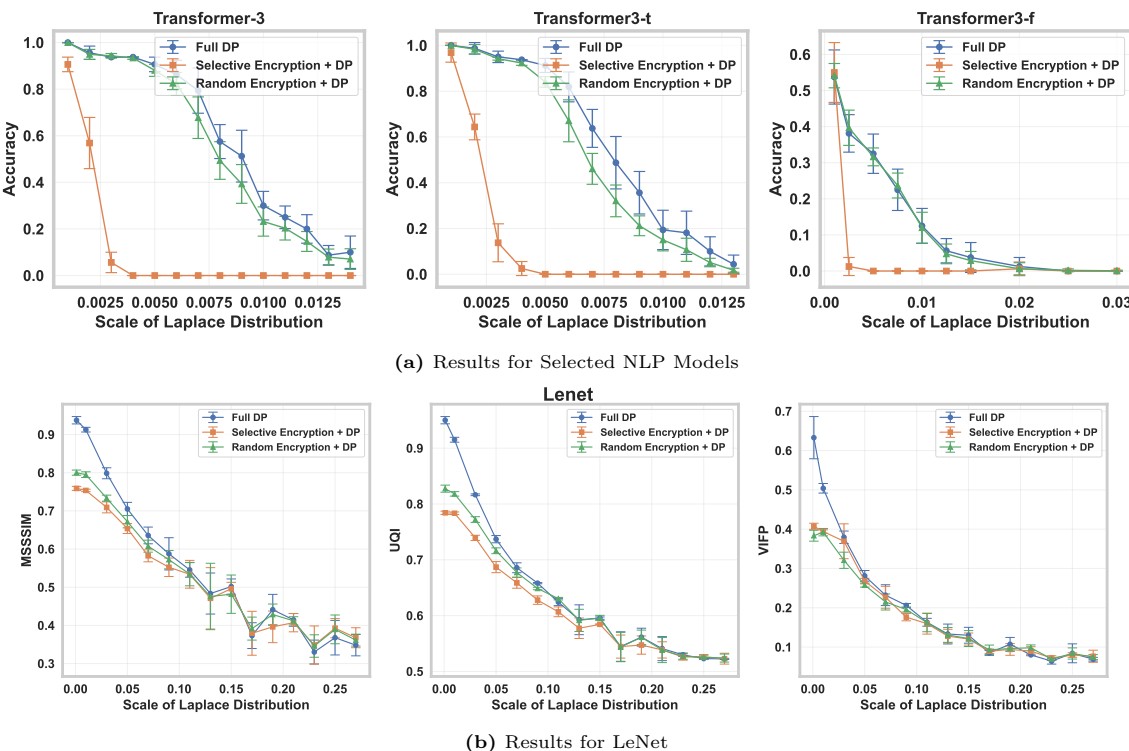

**(a)** Results for Selected NLP Models

**(b)** Results for LeNet

**Figure 23:** Defense Effectiveness of DP Noises of Different Scales Under Three Protection Methods: an encryption ratio is fixed for each model from the beginning to guarantee a good attack performance at first. Each configuration is attacked 10 times and the best attack score is recorded. The experiments are repeated for at least three different sets of applied DP noises.

| Model | Model Size | HE Time (s) | Non-HE Time (s) | Comp Ratio | Ciphertext | Plaintext | Comm Ratio |
|---|---|---|---|---|---|---|---|
| Linear Model | 101 | 0.216 | 0.001 | 150.85 | 266.00 KB | 1.10 KB | 240.83 |
| TimeSeries Transformer | 5,609 | 2.792 | 0.233 | 12.00 | 532.00 KB | 52.65 KB | 10.10 |
| MLP (2 FC) | 79,510 | 0.586 | 0.010 | 60.46 | 5.20 MB | 311.98 KB | 17.05 |
| LeNet | 88,648 | 0.619 | 0.011 | 57.95 | 5.97 MB | 349.52 KB | 17.50 |
| RNN(2 LSTM + 1 FC) | 822,570 | 1.195 | 0.013 | 91.82 | 52.47 MB | 3.14 MB | 16.70 |
| CNN (2 Conv + 2 FC) | 1,663,370 | 2.456 | 0.058 | 42.23 | 103.15 MB | 6.35 MB | 16.66 |
| MobileNet | 3,315,428 | 9.481 | 1.031 | 9.20 | 210.41 MB | 12.79 MB | 16.45 |
| ResNet-18 | 12,556,426 | 19.950 | 1.100 | 18.14 | 796.70 MB | 47.98 MB | 16.61 |
| ResNet-34 | 21,797,672 | 37.555 | 2.925 | 12.84 | 1.35 GB | 83.28 MB | 16.60 |
| ResNet-50 | 25,557,032 | 46.672 | 5.379 | 8.68 | 1.58 GB | 97.79 MB | 16.58 |
| GroupViT | 55,726,609 | 86.098 | 19.921 | 4.32 | 3.45 GB | 212.83 MB | 16.61 |
| Vision Transformer | 86,389,248 | 112.504 | 17.739 | 6.34 | 5.35 GB | 329.62 MB | 16.62 |
| BERT | 109,482,240 | 136.914 | 19.674 | 6.96 | 6.78 GB | 417.72 MB | 16.62 |
| Llama 2 | 6.74 B | 13067.154 | 2423.976 | 5.39 | 417.43 GB | 13.5 GB | 30.92 |

**Table 5:** Vanilla Fully-Encrypted Models of Different Sizes: with 3 clients; Comp Ratio is calculated by time costs of HE over time costs of Non-HE; Comm Ratio is calculated by file sizes of HE over file sizes of Non-HE. CKKS is configured with default crypto parameters.

## A.23    Results on Different Cryptographic Parameters

We evaluate the impacts of variously-configured cryptographic parameters. We primarily look into the packing

| Models | PT (MB) | CT | Opt (MB) |
|---|---|---|---|
| ResNet-18 (12 M) (Tang et al., 2019) | 47.98 | 796.70 MB | 19.03 |
| BERT (110 M) (Hu et al., 2021) | 417.72 | 6.78 GB | 16.66 |

**Table 6:** Parameter Efficiency Overhead: PT means plaintext and CT means ciphertext. Communication reductions are 0.60 and 0.96.

| HE Batch Size | Scaling Bits | Comp (s) | Comm (MB) | Model Test Accuracy $\Delta$ (%) |
|---|---|---|---|---|
| 1024 | 14 | 8.834 | 407.47 | -0.28 |
| 1024 | 20 | 7.524 | 407.47 | -0.21 |
| 1024 | 33 | 7.536 | 407.47 | 0 |
| 1024 | 40 | 7.765 | 407.47 | 0 |
| 1024 | 52 | 7.827 | 407.47 | 0 |
| 2048 | 14 | 3.449 | 204.50 | -0.06 |
| 2048 | 20 | 3.414 | 204.50 | -0.13 |
| 2048 | 33 | 3.499 | 204.50 | 0 |
| 2048 | 40 | 3.621 | 204.50 | 0 |
| 2048 | 52 | 3.676 | 204.50 | 0 |
| 4096 | 14 | 1.837 | 103.15 | -1.85 |
| 4096 | 20 | 1.819 | 103.15 | 0.32 |
| 4096 | 33 | 1.886 | 103.15 | 0 |
| 4096 | 40 | 1.998 | 103.15 | 0 |
| 4096 | 52 | 1.926 | 103.15 | 0 |

**Table 7:** Computational & Communicational Overhead of Different Crypto Parameter Setups: tested with CNN (2 Conv+ 2 FC) and on 3 clients; model test accuracy $\Delta$s is the difference between the best plaintext global model and the best global encrypted global models.

batch size and the scaling bits. The packing batch size determines the number of slots packed in a single ciphertext while the scaling bit number affects the "accuracy" (i.e., how close the decrypted ciphertext result is to the plaintext result) of approximate numbers represented from integers.

From Table 7, the large packing batch sizes in general result in faster computation speeds and smaller overall ciphertext files attributed to the packing mechanism for more efficiency. However, the scaling factor number has an almost negligible impact on overheads.

Unsurprisingly, it aligns with the intuition that the higher bit scaling number results in higher "accuracy" of the decrypted ciphertext value, which generally means the encrypted aggregated model would have a close model test performance to the plaintext aggregated model. However, it is worth mentioning that since CKKS is an approximate scheme with noises, the decrypted aggregated model can yield either positive or negative model test accuracy $\Delta$s, but usually with a negative or nearly zero $\Delta$.

### A.24   Impact from Number of Clients

As real-world systems often experience a dynamic amount of participants within the FL system, we evaluate the overhead shift over the change in the number of clients. Figure 24a breaks down the cost distribution as the number of clients increases. With a growing number of clients, it also means proportionally-added ciphertexts as inputs to the secure aggregation function thus the major impact is cast on the server. When

the server is overloaded, our system also supports client selection to remove certain clients without largely degrading model performance.

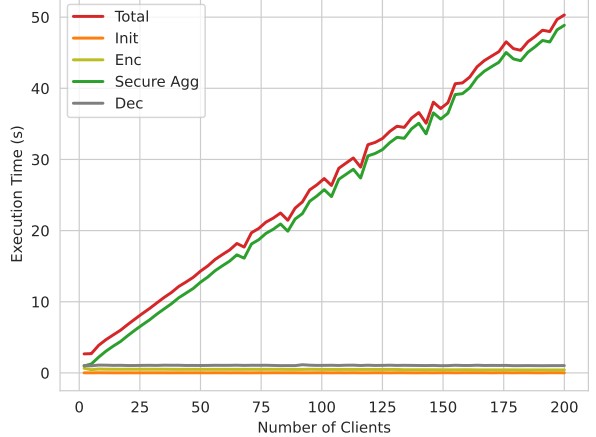

**(a)** Step Breakdown of HE Computational Cost vs. Number of Clients (Up to 200): tested on fully-encrypted CNN

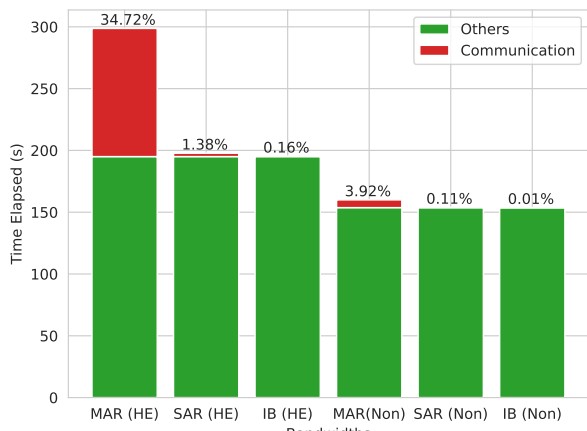

**(b)** Impact of Different Bandwidths on Communication and Training Cycles on Fully-Encrypted ResNet-50: HE means HE-enabled training and Non means plaintext. Others include all other procedures except communication during training. Percentages represent the portion of communication cost in the entire training cycle.

**Figure 24:** Results on Different Number of Clients and Communication Setup

## A.25 Communication Cost on Different Bandwidths

FL parties can be allocated in different geo-locations which might result in communication bottlenecks. Typically, there are two common scenarios: (inter) data centers and (intra) data centers. In this part, we evaluate the impact of the bandwidths on communication costs and how it affects the FL training cycle. We categorize communication bandwidths using 3 cases:

- Infiniband (IB): communication between intra-center parties. 5 GB/s as the test bandwidth.

- Single AWS Region (SAR): communication between inter-center parties but within the same geo-region (within US-WEST). 592 MB/s as the test bandwidth.

- Multiple AWS Region (MAR): communication between inter-center parties but across the different geo-region (between US-WEST and EU-NORTH). 15.6 MB/s as the test bandwidth.

As shown in Figure 24b, we deploy our system on 3 different geo-distributed environments, which are operated under different bandwidths. It is obvious that the secure HE functionality has an enormous impact on low-bandwidth environments while medium-to-high-bandwidth environments suffer limited impact from increased communication overhead during training cycles, compared to Non-HE settings.

## A.26 Different Encryption Selections

Table 8 shows the overhead reductions with different selective encryption rates.

## A.27 Comparison with Other FL-HE Frameworks

Comparison with other popular HE-based FL work can be found in Table 9.

We compare our framework to the other open-sourced FL frameworks with HE capability, namely NVIDIA FLARE (NVIDIA) and IBMFL.

| Selection | Comp (s) | Comm | Comp Ratio | Comm Ratio |
|---|---|---|---|---|
| Enc w/ 0% | 17.739 | 329.62 MB | 1.00 | 1.00 |
| Enc w/ 10% | 30.874 | 844.49 MB | 1.74 | 2.56 |
| Enc w/ 30% | 50.284 | 1.83 GB | 2.83 | 5.69 |
| Enc w/ 50% | 70.167 | 2.83 GB | 3.96 | 8.81 |
| Enc w/ 70% | 88.904 | 3.84 GB | 5.01 | 11.93 |
| Enc w/ All | 112.504 | 5.35 GB | 6.34 | 16.62 |

**Table 8:** Overheads With Different Parameter Selection Configs Tested on Vision Transformer: "Enc w/ 10%" means performs encrypted computation only on 10% of the parameters; all computation and communication results include overheads from plaintext aggregation for the rest of the parameters.

| Features | IBMFL | Nvidia FLARE | Ours |
|---|---|---|---|
| Homomorphic Encryption | ✓ | ✓ | ✓ |
| Threshold Key Management | ✗ | ✗ | ✓ |
| Selective Parameter Encryption | ✗ | ◯ | ✓ |
| Encrypted Foundation Model Training | ◯ | ◯ | ✓ |

**Table 9:** Comparison with Existing HE-Based FL Systems: ◯ implies limited support. For Selective Parameter Encryption, FLARE offers the (random) partial encryption option which does not have clear indications of privacy impacts; for Encrypted Foundation Model Training, the other two platforms require massive resources to train foundation models in encrypted federated learning.

Both NVIDIA and IBMFL utilize Microsoft SEAL as the underlying HE core, with NVIDIA using Open-Minded's python tensor wrapper over SEAL and TenSEAL; IBMFL using IBM'spython wrapper over SEAL and HELayers (HELayers also has an HElib version). Our HE core module can be replaced with different available HE cores, to give a more comprehensive comparison, we also implement a TenSEAL version of our framework for evaluation.

Table 10 demonstrates the performance summary of different frameworks using an example of a CNN model with 3 clients. Our PALISADE-powered framework has the smallest computational overhead due to the performance of the PALISADE library. In terms of communication cost, our system (PALISADE) comes second after IBMFL's smallest file serialization results due to the efficient packing of HELayers' Tile tensors (Aharoni et al., 2011).

Note that NVIDIA's TenSEAL-based realization is faster than the TenSEAL variant of our system. This is because NVIDIA scales each learner's local model parameters locally rather than weighing ciphertexts on the server. This approach reduces the need for the one multiplication operation usually performed during secure aggregation (recall that HE multiplications are expensive). However, such a setup would not suit the scenario where the central server does not want to reveal its weighing mechanism per each individual local model to learners as it reveals partial (even full in some cases) information about participants in the system.

## A.28 Change in Attack Performance over Training

This experiment is used to study the attack performance at different stages of model training. We use Transformer-3 to illustrate the trend as shown in Figure 25. The encryption ratio for random and selective encryption is selected as 0.0005 to guarantee the attack performance at the beginning of the training. The results indicate that the attack performance decreases as the model is trained to be more and more useful, which makes sense since the importance of information contained in the gradient is expected to drop gradually as the training goes toward convergence. Note that the experiment is conducted on only one model because this part is not the main concern of our study. A more comprehensive setup should include multiple CV and NLP models.

| Frameworks | HE Core | Key Management | Comp (s) | Comm (MB) | HE Multi-Party Functionalities |
|---|---|---|---|---|---|
| Ours | PALISADE | ✓ | 2.456 | 105.72 | PRE, ThHE |
| Ours (w/ Opt) | PALISADE | ✓ | 0.874 | 16.37 | PRE, ThHE |
| Ours | SEAL (TenSEAL) | ✓ | 3.989 | 129.75 | — |
| Nvidia FLARE (9a1b226) | SEAL (TenSEAL) | ✓ | 2.826 | 129.75 | — |
| IBMFL (8c8ab11) | SEAL (HELayers) | ◯ | 3.955 | 86.58 | — |
| Plaintext | — | — | 0.058 | 6.35 | — |

**Table 10:** Different Frameworks: tested with CNN (2 Conv + 2 FC) and on 3 clients; Github commit IDs are specified. For key management, our work uses a key authority server; FLARE uses a security content manager; IBMFL currently provides a local simulator.

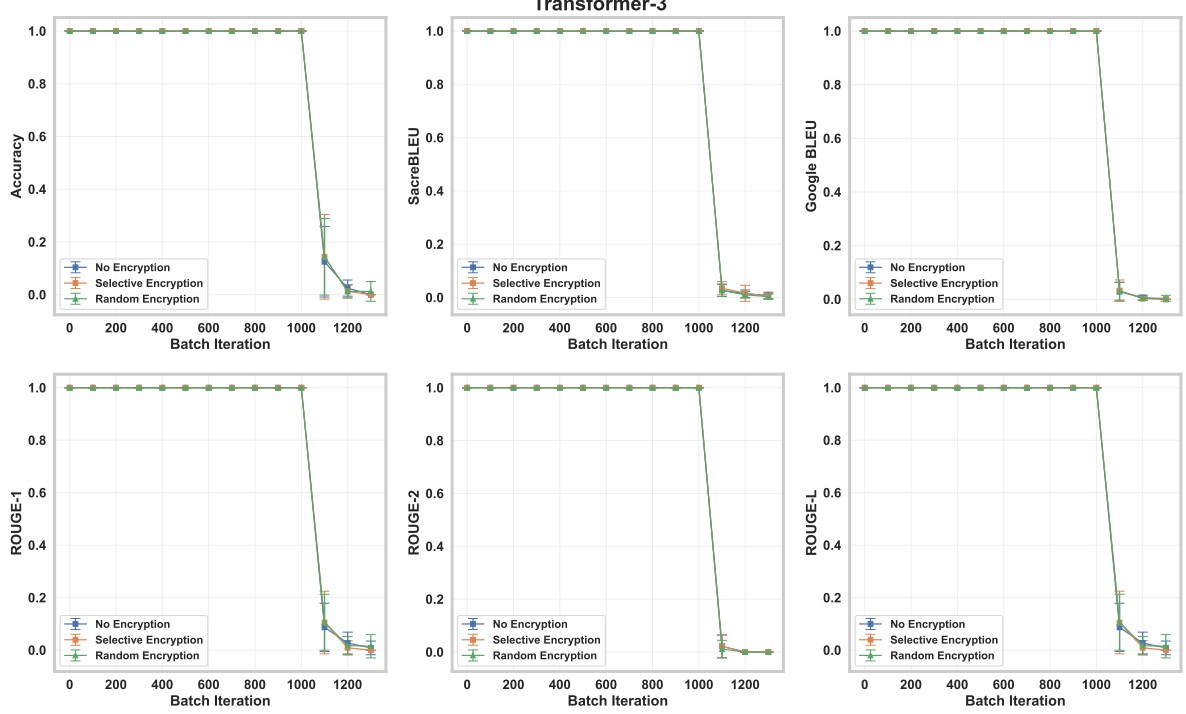

**Figure 25:** Attack Performance on Transformer-3 over Batch Iterations. Each configuration is attacked 10 times and the best score is recorded. The experiment is repeated on 10 different data points and their mean is presented.

## A.29  Overhead Analysis of Parameter Selection

Using the same setup on ResNet-50, we conducted experiments on the overhead introduced by parameter selection to find a selective encryption mask in the initial stage.

The two key steps of parameter selection, namely privacy sensitivity calculation and encrypted global mask agreement, cost 113.8 s and 273.6 s respectively, while the overhead reduction during the entire training task from selective parameter encryption results in 25342.4 s (please refer to the updated Figure for more details) compared to full parameter encryption. This result demonstrates that despite the additional overhead introduced by the parameter selection steps, our method still improves the encrypted FL overheads by a

substantial margin. Additionally, the global mask can be easily reused in different training tasks for the same model architecture with similar data distribution, and the overhead of parameter selection can be further amortized in practice.

## A.30 Client Data Distribution Impact on Sensitivity

Figure 27 shows the difference in sensitivity distribution of Resnet50 under two different client data distributions. The two sensitivity distributions still preserve the characteristics of log-normal mixture distribution, but it is noticeable a slight change in aspects like their mode, range, etc. This observation suggests that alternative global mask aggregation functions, such as maximum-based aggregation, might outperform our current weighted averaging method in terms of privacy protection. It is worth future work to investigate this specific aspect of our selective encryption.

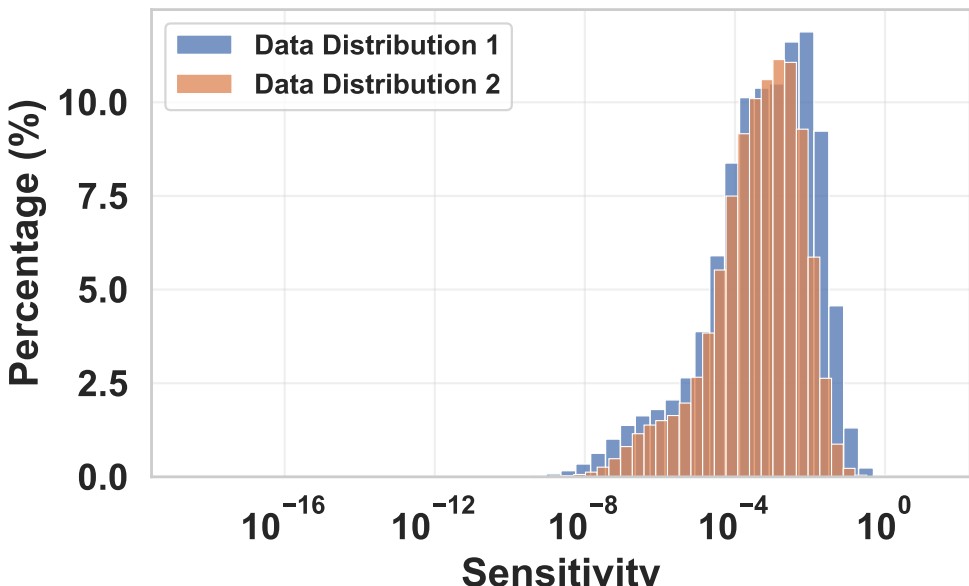

**Figure 27:** Deviation of Sensitivity Distribution Induced by Different Client Data Distribution: two client data distributions constructed from the ImageNet dataset with 100 images from distinct classes sampled at equal intervals. Distribution 1 contains data with labels of [0, 1, 2, 3, 5] while Distribution 2 contains data whose labels span across 0 to 400.

To further investigate this aspect, experimental setups in the previous work Mendieta et al. (2022); Guleria et al. (2024) for the FL data heterogeneity can be considered in future work on this topic regarding privacy sensitivity calculation.

## A.31 Analysis on Newer LLMs

Figure 28 and Figure 29 show how our method performs on newer LLMs from the Llama-3.2 collection. The experimental results indicate that newer LLMs align closely with the findings observed in our experiments on earlier models.

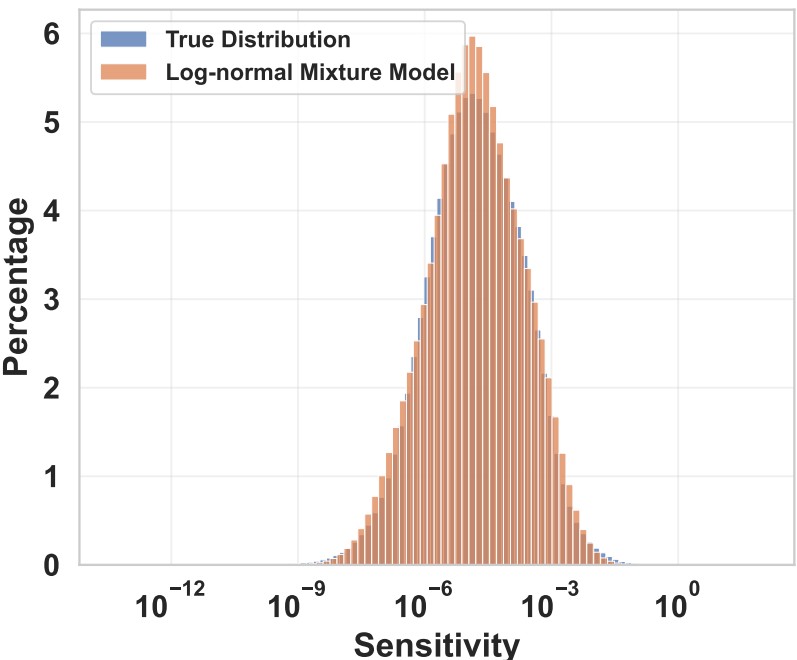

**Figure 28:** Sensitivity Distribution of Llama-3.2-1B

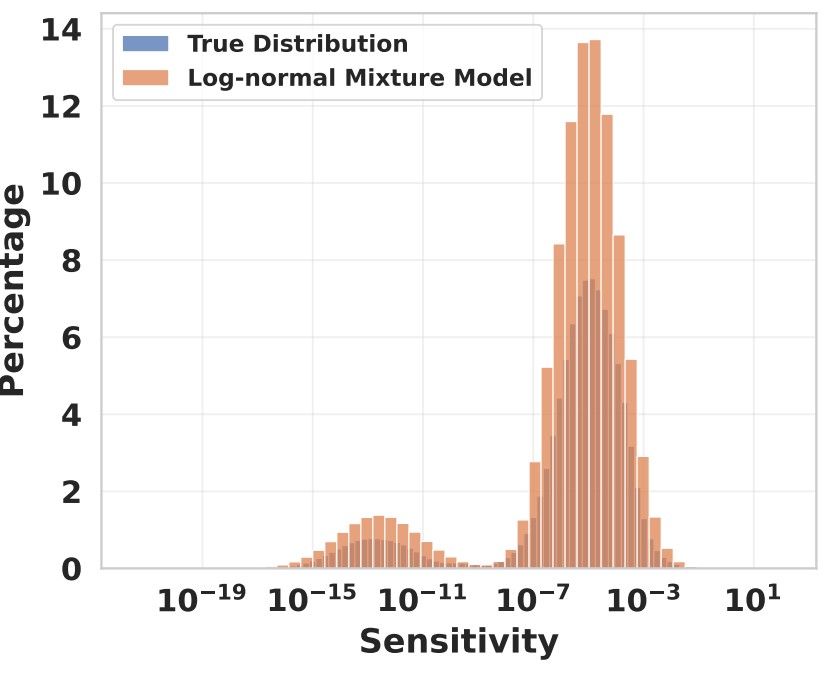

**Figure 29:** Sensitivity Distribution of Llama-3.2-3B

## A.32   MLOps Running Example Configuration

```yaml
common_args:
  training_type: "cross_silo"
  scenario: "horizontal"
  random_seed: 0

data_args:
  dataset: "cifar100"
  partition_method: "hetero"
  partition_alpha: 0.5

model_args:
  model: "resnet50"

train_args:
  federated_optimizer: "FedAvg"
  client_num_in_total: 3
  client_num_per_round: 3
  comm_round: 5
  epochs: 1
  batch_size: 10
  client_optimizer: sgd
  learning_rate: 0.03
  weight_decay: 0.001

validation_args:
  frequency_of_the_test: 5

device_args:
  worker_num: 2
  using_gpu: true
  gpu_mapping_file: config/gpu_mapping.yaml

comm_args:
  backend: "MQTT_S3"
  mqtt_config_path: config/mqtt_config.yaml
  s3_config_path: config/s3_config.yaml

fhe_args:
  enable_fhe: true
  scheme: ckks
  batch_size: 8192
  scaling_factor: 52
  file_loc: "resources/cryptoparams/"

```

**Figure 30:** ResNet-50 MLOps Training Configuration

