# OpenReview forum: "Efficient Privacy-Preserving Federated Learning With Selective Parameter Encryption"
_TMLR — Rejected by TMLR_

### Review · Reviewer_LAwy · 2025-09-02

**Summary Of Contributions:**

This paper introduces a novel privacy-preserving federated learning (FL) framework that uses selective homomorphic encryption (HE) to significantly reduce computational and communication overhead while maintaining strong privacy guarantees. The key contributions are:

1.Selective Parameter Encryption. I think that this is really a novel idea. It encrypts only the most privacy-sensitive model parameters based on a sensitivity map derived from local client data.

2. Theoretical Privacy Framework. The paper gives a formal privacy analysis for selective encryption, showing that it provides a quantifiable privacy budget improvement over full or random encryption.

**Audience:**

Yes

**Claims And Evidence:**

Yes

**Requested Changes:**

I think that this paper is really good. I do not see any important requested changes.

Meanwhile, I suggest to compare with some related works, including:

Jun Feng et al. "Panther: Practical secure 2-party neural network inference." IEEE Transactions on Information Forensics and Security (2025).

**Strengths And Weaknesses:**

Strengths

Novelty: The idea of selectively encrypting only sensitive parameters is intuitive and underexplored, especially with formal privacy guarantees.

Practical Impact: Demonstrated massive overhead reductions (e.g., 100× for GPT-2) make HE-based FL feasible for large models.

Theoretical Rigor: Provides a principled way to quantify privacy under selective encryption using differential privacy and computational indistinguishability.

Comprehensive Evaluation: Covers a wide range of models, datasets, attack scenarios, and system configurations.

Reproducibility: Detailed descriptions of crypto parameters, software architecture, and APIs facilitate replication.

I do not see any important weaknesses.

---

### Review · Reviewer_xaAD · 2025-09-18

**Summary Of Contributions:**

This paper proposes a selective parameter encryption paradigm for PPFL. In particular, only a subset of all parameters are encrypted using the FHE scheme while others are protected by DP noise. Overall, this scheme results in considerable overhead reduction, especially for larger models like GPT-2.

**Audience:**

Yes

**Broader Impact Concerns:**

N/A. I do not find any unaddressed ethical issues.

**Claims And Evidence:**

Yes

**Requested Changes:**

1. Further justify and/or discuss the limiations regarding W1, W2.

2. Fix the formality of the technical terms.

**Strengths And Weaknesses:**

S1. Very significant overhead reduction.

S2. The description is mostly clear and easy to follow.

W1. The privacy guarantee is not rigorous. I understand that the end-to-end privacy leakage is something between the computational zero-leakage (as in FHE) and DP-leakage. However, it is not clear exactly where it is, so I am still skeptical about whether it is providing any improvement in terms of theoretical overhead-privacy tradeoff compared with the vanilla DP-based PPFL. In particular, Step 1 in Section 3.4 is only a **heuristic**.

W2. In addition to W1, the mask $M$ is decided before the training process starts, and is not updated during the training. Accoding to Algorithm 1, it is never updated throughout the entire training process. Therefore, I wonder if the heuristic will still "work" long after the start of training. Meanwhile, if $M$ is indeed updated during the training then it incorporates the information from the training data, which unfortunately costs additional privacy leakage and would further complexify the situation. Therefore, I am not convinced that selective parameter encryption is a viable direction for advancing PPFL.

W3. Please pay attention to the formality of the technical terms. For example, it is more common to use the term "$\epsilon$-DP" or "$\epsilon$-IND-CDP" instead of $\epsilon$-privacy (and please stick to $\epsilon$ for expressing the privacy parameter, unlike Remarks 4.3 and 4.4). Also it may be misleading to mention in Section 4.1 first Gaussian mechanism and then suddenly switch to Laplace mechanism. The authors should more clearly emphasize which mechanism they are using. These presentation issues may confuse the readers who are not very familiar with DP.

---

### Review · Reviewer_nsfQ · 2025-10-01

**Summary Of Contributions:**

This paper proposes a framework for efficient homomorphic-encryption-based federated learning (FL), termed Selective Parameter Encryption (SPE). The idea is to selectively encrypt only the most privacy-sensitive parameters, identified via sensitivity analysis, instead of encrypting all parameters. This approach reduces both computational and communication overhead while preserving provable privacy guarantees. The contributions are:
1. A formal theoretical framework quantifying privacy guarantees of selective encryption.
2. An algorithmic design integrating SPE into HE-based FL pipelines.
3. Experimental validation across CV and NLP tasks (ResNet, LeNet, GPT-2, LLaMA, etc.).

**Audience:**

Yes

**Broader Impact Concerns:**

The work contributes positively by improving privacy in federated learning, but several ethical considerations merit explicit discussion in a Broader Impact Statement. The method assumes semi-honest adversaries and does not address stronger malicious or colluding threat models, leaving residual risks in sensitive applications such as healthcare or finance. Computational costs of homomorphic encryption may exacerbate inequalities in access to privacy-preserving technologies. In addition, sensitivity-based parameter selection could encode biases from local data distributions, raising concerns about fairness and robustness across diverse populations. Finally, while the approach mitigates gradient inversion, other privacy threats such as membership or property inference remain unexplored. These issues suggest the need for a clearer discussion of limitations, fairness, and dual-use implications.

**Claims And Evidence:**

Yes

**Requested Changes:**

1. I was wondering how robust is the sensitivity-based parameter selection under adversarial client manipulation? Could a malicious client skew the global sensitivity map to reduce protection?

2. How does selective encryption interact with advanced FL optimizers beyond FedAvg? is it orthogonal to them? why ?

3. Can the proposed framework handle online/continual training where model architectures evolve, or does the mask need to be recomputed each time?

4. Did you evaluate membership inference attacks, which are relevant for FL privacy?

**Strengths And Weaknesses:**

Strengths:
1. Clear Motivation: The work addresses a very real bottleneck: HE in FL is secure but computationally prohibitive, particularly for foundation models.
2. Novelty: The selective encryption based on sensitivity maps (rather than random masking) is novel and well-justified by both intuition and prior leakage analyses.
3. Theoretical Contribution: The proposed privacy budget analysis formalizes the privacy guarantees of SPE. The link between sensitivity distributions and log-normal mixtures is an insightful observation.
4. Comprehensive Evaluation: Experiments span multiple model families and datasets, include both vision and language tasks, and evaluate against strong inversion attacks. Results convincingly show overhead reduction without sacrificing privacy.

Weaknesses and Concerns:
1. There are some threat model limitations. The adversary model is semi-honest and excludes fully malicious or adaptive adversaries. Many FL systems face stronger threats. Moreover, the multi-key/threshold HE variants are mentioned only in the appendix, but real deployments may require them.

2. The privacy quantification seem to rely heavily on the assumption that parameter sensitivities follow log-normal mixtures. While empirically plausible, the theoretical claims would be stronger with either a general distribution-agnostic bound or robustness checks across atypical distributions.

3. The attack evaluations primarily test gradient inversion and simple language inversion attacks. Stronger privacy attacks (e.g., optimization-based model inversion,  membership inference) are not considered. Furthermore, experiments use relatively small client counts in practice (3–6 clients in many benchmarks). It remains unclear how well the approach scales in cross-device FL with thousands of clients.

4. The paper is dense, with long algorithmic and theoretical sections. Some derivations could be streamlined with intuition before diving into formalism. There are parts of the paper that are difficult to follow.

---

> ### Comment · Reviewer_xaAD · 2025-10-03
> **Further thoughts after reading reviewer nsfQ's reviews**
>
> Thank you, reviewer nsfQ, for your insightful review!
>
> 1. I agree with you on weakness point 2, which may also somehow compromise the theoretical rigorousness and I seemed to have overlooked.
> 2. I also agree that further privacy attacks (like MIA) may be experiemented on, as the theoretically guarantee is not clear due to the use of the heuristics applied.
> 3. However, I would like to defend on behalf of the authors regarding the threat model. I think for studies on FHE, DP, and FL, as well as the intersection of them, it is still common to consider the semi-honest setting. In particular, if you are already using FHE in the computation, it's hard to make sure the servers are doing the computations correctly, since the values are already encrypted. There are indeed pioneering studies on making sure the FHE protocols are executed correctly without compromising the privacy, but they are not often employed in these application-oriented studies for making the protocols efficient and scalable. Having them applied now, in this paper, may be too premature and too demanding. (Please correct me if I misunderstood your point!)
>
> I am also very interested in the authors' response to the reviewer nsfQ's questions in the "requested changes" section, as I believe having them properly addressed will surely increase the clarity of the paper!

---

### Decision · Action_Editor_M6uo · 2025-11-11

**Recommendation:** Reject

**Additional Comments:**

The main idea of this work is to use a mask to selectively encrypt some parameters, in order to reduce the privacy cost caused by noise addition. The mask is generated by an infinitesimal approximation of the sensitivity of each parameter at initialization. In justifying its effectiveness in Theorem 4.5, the authors implicitly assume the mask is accurate (i.e., it masks out the parameters with large sensitivity), which is questionable due to the infinitesimal approximation and the dynamic training of the parameters afterwards. Practically, this does not necessarily cause any issue per se, as one is free to use any mask (even random ones that the authors compared to). It's just harder (if possible) to explicitly quantify the effectiveness. Nevertheless, the writing of the current draft could easily cause confusion: instead of presenting it as an empirical approach to reduce privacy cost, the authors tried to formally quantify their algorithm's effectiveness in Theorem 4.5, which is problematic in several aspects:

- it implicitly assumes the infinitesimal approximation of the sensitivity of each parameter at initialization is accurate, which, as Reviewer xaAD pointed out, is likely false.

- it presents the result in terms of a density function of parameter sensitivities. Since we do not know this density function, the authors had to assume it is log-normal distributed (or some other parametric distribution in the Appendix), which, as Reviewer xaAD and nsfQ both pointed out, breaks the formal DP guarantee. From an experimental evaluation point of view, one can simply compute and compare the savings of the privacy budget for each mask. Why not get rid of Theorem 4.5 and just evaluate the effectiveness of each mask experimentally?

- the experimental evaluation in Section 5.4 is very confusing: it is not clear what roles do r_1 or r_2 play. Without more details it is very hard to see why Table 3 quantifies the privacy guarantees.

Both Reviewer xaAD and nsfQ made a number of excellent suggestions on the presentation and experiments. Regrettably, the authors did not make any changes (or even respond to Reviewer nsfQ). While the proposed selective encryption may seem to hold promise in practice, we believe it is important to address the questions above and the reviewers' comments below.

**Audience:**

Yes

**Audience Explanation:**

The idea to selectively encrypt some parameters to reduce the privacy cost seems interesting to the practitioners of security and privacy.

**Claims And Evidence:**

No

**Claims Explanation:**

The authors implicitly assume that the infinitesimal approximation of the sensitivity in their Algorithm is accurate, based on which they derive Theorem 4.5 to showcase the potential benefits of selective encryption. Another log-normal distributional assumption on the sensitivity is made at the end of Section 4, perhaps only for illustration purpose? It is not clear how the Privacy Guarantee Quantification in Section 5.4 is conducted. The paper appears to be mixing theoretical justifications with empirical shortcuts, with no clear distinction between the two at times and causing unnecessary confusions (as the reviewers pointed out).

**Resubmission Of Major Revision:**

The authors may consider submitting a major revision at a later time.